# MV-SAM: Multi-view Promptable Segmentation using Pointmap Guidance

## Abstract

Promptable segmentation has emerged as a powerful paradigm in computer vision, enabling users to guide models in parsing complex scenes with prompts such as clicks, boxes, or textual cues. Recent advances, exemplified by the Segment Anything Model (SAM), have extended this paradigm to videos and multi-view images. However, the lack of 3D awareness often leads to inconsistent results, necessitating costly per-scene optimization to enforce 3D consistency. In this work, we introduce MV-SAM, a framework for multi-view segmentation that achieves 3D consistency using pointmaps3D points reconstructed from unposed images by recent visual geometry models. Leveraging the pixelpoint one-to-one correspondence of pointmaps, MV-SAM lifts images and prompts into 3D space, eliminating the need for explicit 3D networks or annotated 3D data. Specifically, MV-SAM extends SAM by lifting image embeddings from its pretrained encoder into 3D point embeddings, which are decoded by a transformer using cross-attention with 3D prompt embeddings. This design aligns 2D interactions with 3D geometry, enabling the model to implicitly learn consistent masks across views through 3D positional embeddings. Trained on the SA-1B dataset, our method generalizes well across domains, outperforming SAM2-Video and achieving comparable performance with per-scene optimization baselines on NVOS, SPIn-NeRF, ScanNet++, uCo3D, and DL3DV benchmarks. Code will be released.

## 1 Introduction

Promptable segmentation has emerged as a cornerstone of computer vision, enabling humans to efficiently guide models in parsing complex scenes through simple prompts such as clicks, boxes, or textual cues. The success of the Segment Anything Model (SAM) (Kirillov et al., 2023) has demonstrated the potential of this paradigm across a wide range of 2D applications. Building on SAM, follow-up works (Cheng & Schwing, 2022; Bekuzarov et al., 2023; Cheng et al., 2024) extend promptable segmentation to videos and multi-view images to achieve consistent scene-level segmentation. Most notably, SAM2 (Ravi et al., 2024) introduces a mask propagation mechanism based on memory-attention layers, achieving strong cross-domain generalization. Despite these advances, methods relying on temporal continuity inherently lack 3D awareness, making them prone to occlusion, object reappearance, or repetitive visual patterns.

A key challenge in incorporating 3D awareness into segmentation models lies in designing a 3D representation that can naturally operate with 2D user prompts. In detail, conventional 3D representationssuch as point clouds, meshes, voxel grids, or 3D Gaussians (Kerbl et al., 2023)decouple 3D geometry from 2D imagery. Consequently, mapping 2D user prompts (e.g., a mouse click on an image) into the corresponding 3D position typically requires rendering or projection steps, which introduce unnecessary complexity and remain susceptible to occlusion.

Recent advances in visual geometry models (Wang et al., 2025a;c) provide a new opportunity. Visual Geometry Grounded Transformer (VGGT) (Wang et al., 2025a) and its follow-up work $\pi^3$ (Wang et al., 2025c) have demonstrated that dense 3D reconstructions can be obtained directly from unposed images in a single feed-forward manner while bypassing conventional structure-from-motion pipelines (Schönberger & Frahm, 2016; Schönberger et al., 2016). Resulting reconstructions, termed *pointmap*, are geometrically faithful and preserve a strict one-to-one correspondence between pixels

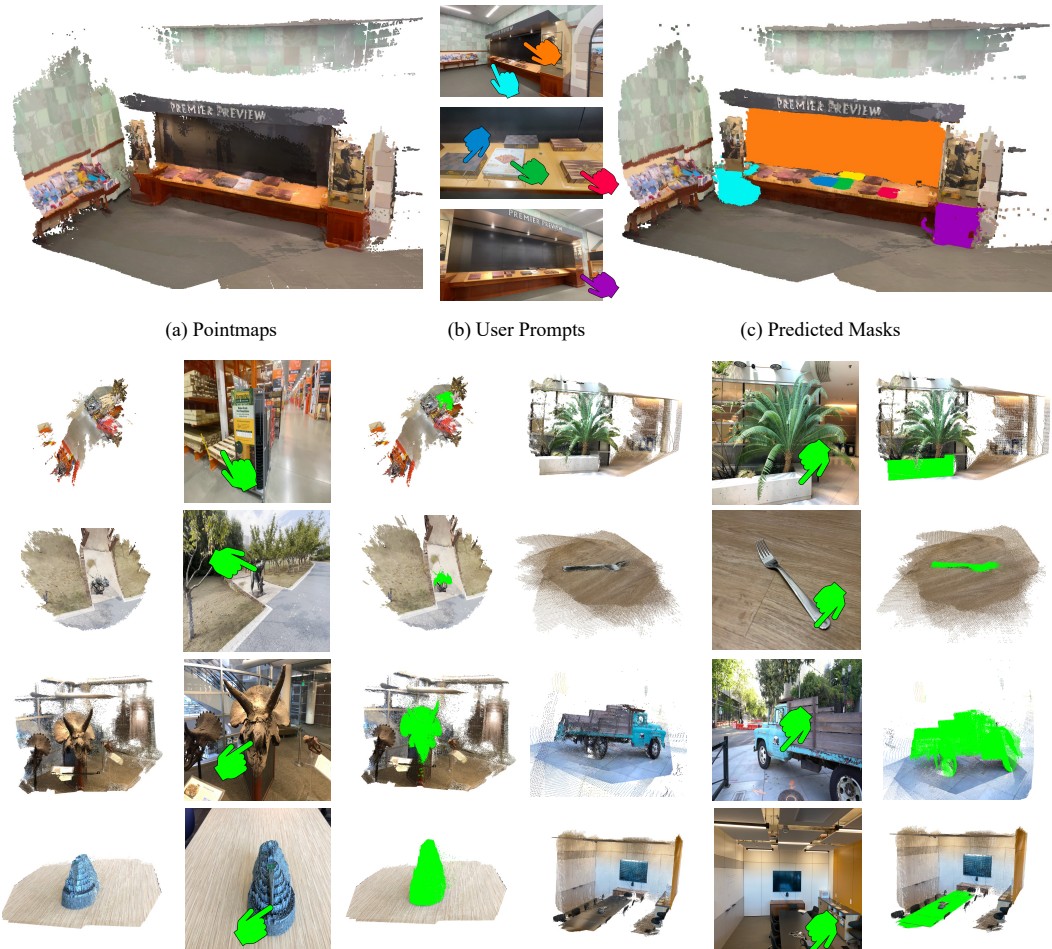

(a) Pointmaps      (b) User Prompts      (c) Predicted Masks

Figure 1: MV-SAM. Our method enables view-consistent promptable segmentation, where user prompts (e.g., clicks) guide the extraction of target masks across multi-view images. The examples above illustrate visualizations of the lifted predicted masks via pointmaps, where finger points (illustrating user prompts) share the same color as the corresponding predicted masks.

and points. This property forms a natural bridge between 2D prompts and 3D geometry, eliminating the need for rendering or projection.

Building on this insight, we propose **MV-SAM**, a multi-view promptable segmentation framework that operates on pointmaps reconstructed from unposed images by the visual geometry model. By leveraging the strict one-to-one pixel-point correspondence, we seamlessly transfer pretrained 2D segmentation knowledge (e.g., from SAM) into 3D space without relying on specialized 3D architectures or large-scale annotated 3D datasets. Our framework is threefold: (1) reusing the SAM2-Video's image encoder to extract rich image embeddings, (2) embedding 3D positions on both image embeddings and prompts, and (3) decoding masks using a lightweight transformer network.

We evaluate MV-SAM on various benchmarks: NVOS (Ren et al., 2022), SpIn-NeRF (Mirzaei et al., 2023), ScanNet++ (Yeshwanth et al., 2023), uCo3D (Liu et al., 2025a), and DL3DV (Ling et al., 2024) as visualized in Figure 1. Our MV-SAM consistently outperforms the image/video-based foundation model SAM2-Video (Ravi et al., 2024), and achieves competitive performance against optimization-based baselines (Cen et al., 2023b; 2025; 2023a; Ying et al., 2024). In summary, our contributions are threefold:

- We address multi-view promptable segmentation using pointmaps3D points reconstructed from unposed imageswhich eliminate the need for rendering or projection to align 2D interactions with 3D geometry.

- We leverage the one-to-one pixelpoint correspondence of pointmaps to lift both user prompts and image embeddings from the pretrained SAM2-Video encoder into 3D space, thereby enabling our model to directly transfer rich 2D knowledge into 3D understanding.

- We demonstrate that a lightweight transformer network with 3D positional embeddings is sufficient for robust, view-consistent, and 3D-aware promptable segmentation, eliminating the need for specialized 3D networks or mask-annotated 3D data.

## 2 RELATED WORK

**Promptable segmentation.** Promptable segmentation aims to predict binary masks conditioned on both input modality and user inputs such as clicks, boxes, or masks. The Segment Anything Model (SAM) (Kirillov et al., 2023) popularized this task by introducing a large-scale, prompt-driven image segmenter capable of generalizing across a wide range of domains and capture conditions. Its extensions incorporate temporal cues to handle videos, either by combining SAM with trackers (Cheng et al., 2023a; Yang et al., 2023; Cheng et al., 2023b; Rajič et al., 2023) or by introducing memory mechanisms as in SAM2 (Ravi et al., 2024). While highly effective in 2D, these models lack explicit 3D understanding, which often leads to predict inconsistent masks across views. Our work addresses this limitation by directly linking 2D prompts to 3D geometry through pointmaps.

**Multi-view segmentation.** Multi-view segmentation aims to achieve consistent segmentation across multiple viewpoints. Prior work often enforces this consistency through explicit 3D data representations or specialized 3D neural networks. Some approaches optimize volumetric representations with geometric constraints (Ren et al., 2022), while others leverage neural fields (Mildenhall et al., 2021; Kerbl et al., 2023) to produce consistent multi-view masks (Mirzaei et al., 2023). Another line of work lifts 2D segmentations into 3D, either by fusing SAM predictions into pointclouds (Cen et al., 2023b) or by learning contrastive 3D feature spaces (Ying et al., 2024). More recently, 3D Gaussian Splatting method (Kerbl et al., 2023) have been explored for interactive and promptable segmentation, where Gaussians are augmented with features (Choi et al., 2025), affinities (Cen et al., 2023a), or propagation strategies (Sun et al., 2025). While effective, these methods rely on heavy per-scene optimization or computationally demanding pipelines. In parallel, SAMPro3D (Xu et al., 2025) introduces 3D promptable segmentation by leveraging 2D3D correspondences, but its reliance on pre-existing 3D data limits applicability to diverse domains. In contrast, our framework achieves strong generalization without explicit 3D supervision or scene-specific optimization.

**3D reconstruction from unposed images.** Reconstructing 3D structures from images has long been a central challenge in computer vision. Traditional Structure-from-Motion (Schönberger & Frahm, 2016; Schönberger et al., 2016; Pan et al., 2024) relies on bundle adjustment and iterative optimization, thereby requiring extensive optimization time. To improve efficiency, recent studies have proposed feed-forward predictors of pointmaps (Wang et al., 2024; Leroy et al., 2024), which were subsequently extended with global alignment strategies (Wang & Agapito, 2024; Wang et al., 2025b). Recent work such as FASt3R (Yang et al., 2025) and FLARE (Zhang et al., 2025) introduced feed-forward architectures for jointly predicting camera poses and 3D points. VGGT (Wang et al., 2025a) advanced this line by fully predicting dense pointmaps, camera poses, depth, and point-tracking features, demonstrating strong cross-domain generalization. Building on this, $\pi^3$ (Wang et al., 2025c) employs permutation-equivariant transformers to address order sensitivity, enabling robust reconstructions in both static and dynamic scenes. In this work, we leverage the pointmap predictor $\pi^3$ to estimate 3D positions of image pixels and user prompts, enabling 2D prompts to be seamlessly transferred into 3D space.

**3D network architecture.** To process sparse 3D data, 3D vision researchers have explored network architectures capable of operating on unordered point sets. As pioneers in this direction, PointNet Qi et al. (2017a) and PointNet++ Qi et al. (2017b) introduce MLP-based architectures that directly consume a fixed number of points. However, applying pure MLPs to scene-level point clouds leads to prohibitive memory consumption. Consequently, follow-up work such as MinkowskiNet (Choy et al., 2019) proposes memory-efficient sparse convolutions on voxelized representations. More recently, inspired by the success of transformers in various vision domains, PTv3 (Wu et al., 2024) introduces a transformer-based architecture that achieves state-of-the-art performance across multiple benchmarks.

(a) SAM2-Video

(b) MV-SAM (Ours)

Figure 2: Overview. (a) SAM2-Video tracks masks iterative through memory modules where the visual cues are the key for tracking. In contrast, (b) our MV-SAM leverage pointmap as a unified world coordinate and embed 3D positional information in both image embeddings and user prompts to predict view-consistent masks without 3D-specific networks or large-scale 3D annotated datasets.

Despite these advances, contemporary 3D architectures still rely heavily on metric-depth aligned point clouds and often struggle to generalize to unseen domains. In this work, we demonstrate that removing explicit 3D operations from MV-SAM leads to significantly stronger generalization performance.

## 3 METHODOLOGY

Multi-view promptable segmentation aims to localize all corresponding regions from unposed images of a scene using seed user prompts on the images. Let us denote the collection of $N$ unposed images by $\mathcal{I} = \{I_i\}_{i=1}^N$ and the set of seed prompts by $\mathcal{S} = \{S_i\}_{i=1}^{N_S}$ given to $N_S$ images among them ($N_S < N$) such that the input can be divided into a prompted set $\mathcal{I}_S = \{(I_i, S_i)\}_{i=1}^{N_S}$ and a unprompted set $\mathcal{I}_U = \{I_i\}_{i=N_S+1}^N$. Then, the task is defined to predict the set of object/part masks $\hat{\mathcal{M}} = \{\hat{M}_i\}_{i=1}^N$ for all the images, corresponding to the given prompts $\mathcal{S}$.

In Figure 2, we provide an overview of our MV-SAM. Our framework consists of three stages: (1) a pre-processing stage that reconstructs pointmaps from unposed images using $\pi^3$ (Wang et al., 2025c) and extracts image embeddings using the pretrained image encoder of SAM2-Video (Ravi et al., 2024) (Section 3.1), (2) a positional embedding stage that locates image embeddings and user seed prompts on 3D and embed 3D positional information(Section 3.2), and (3) a mask decoding stage that predicts view-consistent masks (Section 3.3).

### 3.1 PRE-PROCESSING STAGE

In this stage, we process a set of unposed images $\mathcal{I} = \{I_i\}_{i=1}^N$ to obtain pointmaps $\mathcal{P} = \{P_i\}_{i=1}^N$ and corresponding image embeddings $\mathcal{F} = \{F_i\}_{i=1}^N$. We begin by applying an off-the-shelf visual geometry model, $\pi^3$ (Wang et al., 2025c), to reconstruct a pointmap $P_i = [\mathbf{p}_{ip}]_{p=1}^{N_P}$, where $N_P$ denotes the number of points in the pointmap and each 3D point is represented as $\mathbf{p}_{ip} \in \mathbb{R}^3$. For detailed preliminaries about $\pi^3$, please refer to Appendix A.2. The point $\mathbf{p}$ has strict one-to-one correspondence with an image pixel $\mathbf{r}_{ip} \in \mathbb{R}^2$, eliminating the need for rendering or projection to bridge 2D and 3D data. In addition to 3D coordinates, the $\pi^3$ predicts a set of confidence maps $\mathcal{C} = \{C_i\}_{i=1}^N$, where each confidence map $C_i$ consists of the value $c_{ip} \in \mathbb{R}$ that indicates the recon-

struction reliability of corresponding $\mathbf{p}_{ip}$. During this stage, we also obtain 3D prompts, $\{S_i^{3\mathrm{D}}\}_{i=1}^{N_s}$, by mapping $\mathcal{S}$ into their corresponding 3D points using $\mathcal{P}$.

## 3.2 3D POSITIONAL EMBEDDING FOR USER PROMPTS AND IMAGE EMBEDDINGS

Previous promptable segmentation models interpret user prompts within images or points to predict the indicated regions. Among them, SAM2-Video (Figure 2a) relies on 2D positional embeddings, assigned independently to each frame, together with memory modules that propagate masks and prompts across frames. For multi-view promptable segmentation (Figure 2b), we instead embed both user prompts and image features with 3D positional encodings, defined in a common world coordinate system derived from pointmaps $\mathcal{P}$, to enable consistent mask prediction across views.

**3D positional embeddings from pointmaps.** We observe that the extracted pointmaps exhibit varying standard deviations depending on the number of frames, which brings various scene scales. To address this, we empirically found that applying standardization across all points yields consistent predictions that remain robust under changes in the number of frames. For empirical details, please refer to Appendix E. Specifically, we compute the mean and standard deviation over all points $\mathcal{P}$ and apply z-score normalization to enforce a unit standard deviation and zero mean. The standardized coordinates $\tilde{\mathbf{p}}_{ip} \in \tilde{P}_i$ are then passed through the sinusoidal positional embedding, which is computed as:

$$\mathbf{f}_{ip}^{\mathrm{PE}} = [\sin(2\pi\mathbf{b}^\top\tilde{\mathbf{p}}_{ip}),\ \cos(2\pi\mathbf{b}^\top\tilde{\mathbf{p}}_{ip})]^\top \ \ s.t. \ \tilde{\mathbf{p}}_{ip} = \frac{\mathbf{p}_{ip} - \boldsymbol{\mu}}{\boldsymbol{\sigma}}, \tag{1}$$

where $\boldsymbol{\mu} \in \mathbb{R}^3$ is the mean, $\boldsymbol{\sigma} \in \mathbb{R}^3$ is the standard deviation, $\mathbf{f}_{ip}^{\mathrm{PE}} \in F_i^{\mathrm{PE}}$ is a positional embedding vector, and $\mathbf{b} \in \mathbb{R}^{3 \times 64}$ is the Fourier basis frequency (Tancik et al., 2020; Kirillov et al., 2023).

**Prompt embeddings.** We mostly follow the prompt encoder design introduced by SAM (Kirillov et al., 2023) except for the positional embeddings. We use 3D positional embeddings instead of 2D. Then the formulation to compute prompt embedding from a 3D prompt $\mathbf{s}_{ip}^{3\mathrm{D}} \in S_i^{3\mathrm{D}}$ is:

$$\mathbf{s}_{ip}^{\mathrm{PE}} = \begin{cases} \mathbf{f}_{ip}^{\mathrm{PE}} + \mathbf{f}^{\mathrm{pos}}, & i \in [1, N_{\mathrm{S}}] \text{ and } \mathbf{s}_{ip}^{3\mathrm{D}} \text{ is a positive prompt,} \\ \mathbf{f}_{ip}^{\mathrm{PE}} + \mathbf{f}^{\mathrm{neg}}, & i \in [1, N_{\mathrm{S}}] \text{ and } \mathbf{s}_{ip}^{3\mathrm{D}} \text{ is a negative prompt,} \end{cases} \tag{2}$$

where $\mathbf{f}^{\mathrm{pos}} \in \mathbb{R}^{128}$ and $\mathbf{f}^{\mathrm{neg}} \in \mathbb{R}^{128}$ are learnable embeddings corresponding to positive and negative prompt, respectively.

**Confidence embeddings.** While $\pi^3$ generally provides accurate reconstructions, points with low confidence can still be inaccurately localized, and incorporating such points as prompts adversely affects segmentation performance. To mitigate this issue, we introduce two learnable embeddings: one for high-confidence points and another for low-confidence points. These embeddings are added to the positional embeddings of both user prompts and pointmaps, allowing the model to modulate its attention according to the confidence score maps $\mathcal{C}$. We define low-confidence points as the bottom 15% of all points ranked by confidence, denoted by $c^{\mathrm{th}}$. For each pixel coordinate $\mathbf{r}$ and each seed prompt $\mathbf{s}$, we then define their positional embeddings as follows::

$$\hat{\mathbf{f}}_{ip}^{\mathrm{PE}} = \begin{cases} \mathbf{f}_{ip}^{\mathrm{PE}} + \mathbf{f}^{\mathrm{hc}}, & c_{ip} > c^{\mathrm{th}}, \\ \mathbf{f}_{ip}^{\mathrm{PE}} + \mathbf{f}^{\mathrm{lc}}, & \text{else,} \end{cases} \qquad \hat{\mathbf{s}}_{ip}^{\mathrm{PE}} = \begin{cases} \mathbf{s}_{ip}^{\mathrm{PE}} + \mathbf{f}^{\mathrm{hc}}, & c_{ip} > c^{\mathrm{th}}, \\ \mathbf{s}_{ip}^{\mathrm{PE}} + \mathbf{f}^{\mathrm{lc}}, & \text{else,} \end{cases} \tag{3}$$

where $\mathbf{f}^{\mathrm{hc}} \in \mathbb{R}^{128}$ and $\mathbf{f}^{\mathrm{lc}} \in \mathbb{R}^{128}$ are learnable embeddings corresponding to high-confidence and low-confidence score, respectively. $c^{\mathrm{th}} \in \mathbb{R}$ is a threshold for the confidence. In our experiments, we dynamically set the threshold $c^{\mathrm{th}}$ by selecting the lowest 15% of confidence scores across all views.

**Point embeddings.** Given an image embedding vector $\mathbf{f}_{ip} \in F_i$ and a 3D positional embedding vector $\hat{\mathbf{f}}_{ip}^{\mathrm{PE}} \in \hat{F}_i^{\mathrm{PE}}$, we calculate a point embedding $\hat{\mathbf{f}}_{ip}^{\mathrm{PE}} \in \hat{F}_i^{\mathrm{PE}}$ by $\hat{\mathbf{f}}_{ip}^{\mathrm{PE}} = \mathbf{f}_{ip} + \hat{\mathbf{f}}_{ip}^{\mathrm{PE}}$.

## 3.3 MASK DECODER

Given point embeddings $\hat{\mathcal{F}}^{\mathcal{P}}$ and prompt embeddings $\hat{\mathcal{S}}^{\mathrm{PE}}$, our mask decoder is trained to predict masks $\hat{\mathcal{M}} = \{\hat{M}_i\}_{i=1}^N$. We adopt the two-way transformer design of SAM2-Video, which restricts the attention scope to individual viewpoints. Specifically, SAM2-Video applies independent 2D

Table 1: Comparison of SAM2-Video and our MV-SAM under different prompt settings using three benchmarks: ScanNet++, uCo3D, and DL3DV. Note that we have slightly improved the performance of MV-SAM by training for more epochs without modifying any hyperparameters. In addition, we refined the manually annotated masks in DL3DV, which resulted in updated performance.

| Dataset | Video | | Multi-view Images | |
|---|---|---|---|---|
| | SAM2-Video mIoU (↑) / mAcc (↑) | MV-SAM mIoU (↑) / mAcc (↑) | SAM2-Video mIoU (↑) / mAcc (↑) | MV-SAM mIoU (↑) / mAcc (↑) |
| ScanNet++ | 46.1 / 61.4 | **48.9 / 63.5** | 47.5 / 62.8 | **49.1 / 62.9** |
| uCo3D | 81.9 / 91.3 | **87.7 / 95.0** | 83.2 / 91.9 | **87.4 / 95.1** |
| DL3DV | 67.3 / 82.9 | **75.1 / 91.8** | 64.2 / 78.6 | **75.0 / 92.0** |
| Average | 65.1 / 78.5 | **70.6 / 83.4** | 65.0 / 77.8 | **70.5 / 83.3** |

positional embeddings to each frame and uses memory attention to implicitly track masks from previous frames. In contrast, our mask decoder utilizes 3D positional embeddings to place all image pixels within a unified world coordinate system defined by pointmaps. The primary difference between the two methods is that SAM2-Video propagates predicted masks to subsequent frames via a memory attention layer, whereas MV-SAM inherently propagates prompts across all frames using 3D positional embeddings. Accordingly, our mask decoder is able to locate all the possible prompts $\hat{\mathcal{S}}^{\text{PE}}$ for every $\hat{F}_i^{\mathcal{P}}$ through transformer layers to predict masks $\hat{M}_i$ (referred to as 'single-view' in Table 3a). Specifically, our decoder takes $\hat{F}_i^{\mathcal{P}}$ as queries and $\hat{\mathcal{S}}^{\text{PE}}$ as keys and values, and the output mask $\hat{M}_i$ is formulated as:

$$\hat{M}_i = \text{Decoder}(\hat{F}_i^{\mathcal{P}}, \hat{\mathcal{S}}^{\text{PE}}). \tag{4}$$

Our architecture is fully equivariant to frame order, since $\pi^3$ is designed to satisfy permutation equivariance. This property ensures that the model yields consistent performance even when the input frames are randomly permuted, as shown in Table 1.

## 3.4 TRAINING LOSS

We train our model using a standard binary segmentation loss that supervises the predicted masks against ground truth object masks. For scalability, we train our model on SA-1B Kirillov et al. (2023), a large-scale dataset with mask annotations of objects. Importantly, our model is trained solely on single-view objectimage pairs, without requiring any multi-view datasets to achieve multi-view consistency. Specifically, we employ a combination of focal loss (Lin et al., 2017) and dice loss (Milletari et al., 2016) to handle class imbalance and ensure sharp mask boundaries.

Given ground-truth 2D masks $\mathcal{M}$, we first randomly sample sparse or dense prompts from $\mathcal{M}$. Then, our method infers predicted masks $\hat{\mathcal{M}}$ from unposed images $\mathcal{I}$ and the prompts $\mathcal{S}$ through the frozen image encoder $\theta^*_{\text{imgenc}}$, trainable parameters $\theta$ such as the mask decoder $\theta_{\text{dec}}$, the prompt encoder $\theta_{\text{penc}}$, and the confidence embeddings $\theta_{\text{conf}}$, which is optimized by minimizing the losses:

$$\min_{\theta} \mathcal{L} = \min_{\theta} \left( \lambda_{\text{focal}} \mathcal{L}_{\text{focal}} + \lambda_{\text{dice}} \mathcal{L}_{\text{dice}} \right), \tag{5}$$

Detailed hyperparameters are described in Section A of the Appendix.

## 4 EXPERIMENTS

### 4.1 PROMPTABLE SEGMENTATION ON VIDEOS OR MULTI-VIEW IMAGES

We compare MV-SAM with SAM2-Video across diverse domains to assess the generalizable capability of our model. The evaluation covers multiple benchmarks: real-world object-centric scenes (Liu et al., 2025a), outdoor scenarios (Ling et al., 2024), and indoor environments (Yeshwanth et al., 2023). To reflect practical scenarios, we consider two evaluation settings: (1) Video: temporally coherent video inputs, and (2) MV-Images: video frames randomly permuted to simulate multi-view image inputs. For evaluation, we randomly sample 10 positive prompts and 2 negative prompts from the target masks across different viewpoints. Positive prompts are sampled

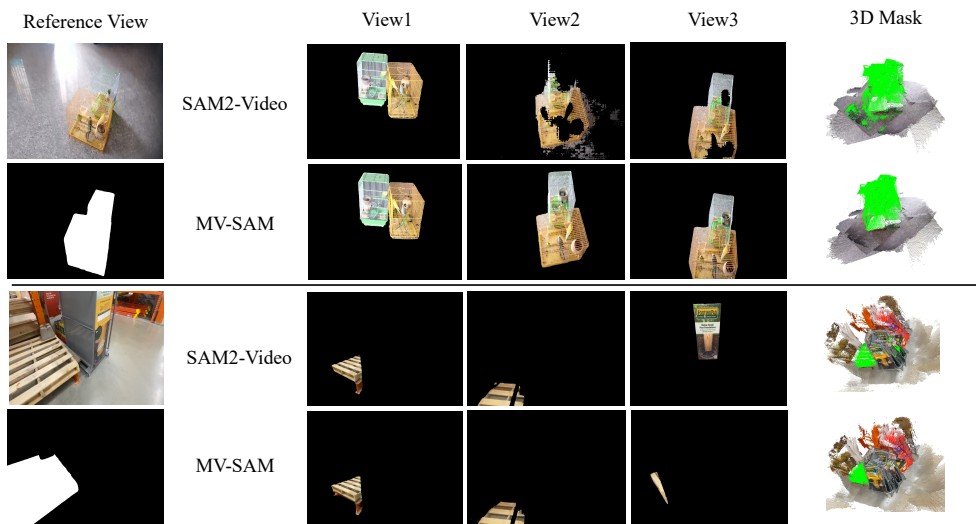

Figure 3: Comparison of MV-SAM with SAM2-Video.

Table 2: Comparison on NVOS and SPIn-NeRF benchmarks. Note that we have slightly improved the performance of MV-SAM by training for more epochs without modifying any hyperparameters.

| Category | Method | NVOS | | SPIn-NeRF | |
|---|---|---|---|---|---|
| | | mIoU (↑) | mAcc (↑) | mIoU (↑) | mAcc (↑) |
| per-scene opt. | SPIn-NeRF (Mirzaei et al., 2023) | - | - | 90.7 | 98.8 |
| | SA3D (Cen et al., 2023b) | 91.1 | 98.4 | 92.4 | 98.8 |
| | SAGA (Cen et al., 2023a) | 92.6 | 98.6 | 93.7 | 99.2 |
| | SA3D-GS (Cen et al., 2025) | 92.7 | 98.5 | 93.4 | 99.1 |
| | OmniSeg3D (Ying et al., 2024) | 92.8 | 98.6 | 94.5 | 99.3 |
| generalization | SAM2-Video (Ravi et al., 2024) | 88.7 | 94.6 | 86.6 | 93.6 |
| | MV-SAM (ours) | **92.1** | **97.5** | **92.9** | **97.1** |

from within the target region, whereas negative prompts are sampled from non-target regions. For fairness, we use the same set of prompts when comparing the models.

Table 1 showcases the superiority of MV-SAM over SAM2-Video across different domains. Our MV-SAM consistently outperforms SAM2-Video across diverse datasets in both video and multi-view image settings. As illustrated in the first example of Figure 3, SAM2-Video often fails to reliably track objects, frequently introducing holes or missing object parts during mask propagation.

### 4.2 MULTI-VIEW SEGMENTATION: NVOS AND SPIN-NERF

We further evaluate MV-SAM, SAM2-Video and prior per-scene optimization methods (NeRF-based approaches (Mirzaei et al., 2023; Ying et al., 2024), Gaussian-based approaches (Cen et al., 2023a; 2025), and depth-based methods Cen et al. (2023b)) on the NVOS Ren et al. (2022) and SPIn-NeRF Mirzaei et al. (2023) benchmarks, which are widely used for evaluating multi-view promptable segmentation performance[1].

The datasets consist of multi-view images without guaranteed temporal consistency across frames. Following the evaluation protocol of SAGA (Cen et al., 2023a), we randomly sample 8 positive prompts and 2 negative prompts. For NVOS, the prompts are derived from scribbles, while for SPIn-NeRF they are sampled from reference-view masks.

As shown in Table 2, our method outperforms SAM2-Video, while performing competitive with per-scene optimization approaches. However, the heavy computational cost of per-scene optimiza-

---

[1]Note that we exclude the 'orchid' scene in NVOS and the 'pinecone' scene in SpIn-NeRF; see the Appendix Section A for reasoning and details.

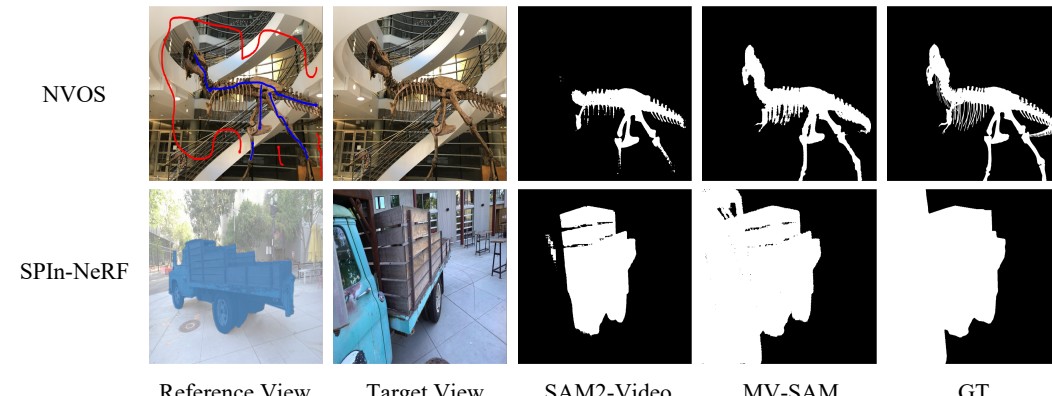

| | Reference View | Target View | SAM2-Video | MV-SAM | GT |

Figure 4: Qualitative results on NVOS and SPIn-NeRF datasets. Compared to SAM2-Video, which often predicts incorrect regions due to the lack of 3D awareness, MV-SAM achieves consistent mask predictions. In NVOS, user prompts are provided as scribbles, where blue lines visualize positive scribbles and red lines visualize negative scribbles, whereas in SPIn-NeRF, object masks in the reference view serve as user prompts.

Table 3: Ablation study of (a) the mask decoder network and (b) the encoder network. CP denotes the usage of confidence embeddings, PE refers to positional embeddings, Norm indicates normalization of 3D positions as described in section 3.1, and Attn abbreviates the attention operator. We use MinkUnet42 (37.9M) (Choy et al., 2019) for 3D encoder baseline. '3D rep. stands for 3D representation where we introduce residual blocks for 2D-to-3D conversion by adopting sparse voxel representation (Choy et al., 2019) and point representation (Wu et al., 2024).

(a) Mask decoder network.

| CP | PE | Attn | mIoU (↑) | mAcc (↑) |
|---|---|---|---|---|
| | 3D PE | single-view | 44.5 | 61.1 |
| ✓ | 3D PE | single-view | **52.2** | **66.7** |
| ✓ | No PE | full-view | 25.6 | 57.8 |
| ✓ | No PE | single-view | 10.9 | 52.7 |
| ✓ | 2D PE | full-view | 26.6 | 59.5 |
| ✓ | 2D PE | single-view | 18.3 | 53.6 |
| ✓ | 3D PE | full-view | 45.8 | 62.2 |
| ✓ | 3D PE | single-view | **52.2** | **66.7** |

(b) Encoder network.

| Encoder | 3D rep. | Grid Size | mIoU (↑) | mAcc (↑) |
|---|---|---|---|---|
| 3D encoder (Mink) | Voxel (Mink) | 0.005 | 37.2 | 64.4 |
| Image encoder (SAM2) | Voxel (Mink) | 0.05 | 40.6 | 63.5 |
| | | 0.01 | 41.3 | 65.0 |
| | | 0.005 | 44.3 | 64.5 |
| Image encoder (SAM2) | Voxel (PTv3) | 0.05 | 40.7 | 64.3 |
| | | 0.01 | 41.0 | 65.2 |
| | | 0.005 | 42.1 | 66.3 |
| Image encoder (SAM2) | 3DPE (ours) | - | **52.2** | **66.7** |

tion makes those methods impractical for interactive graphics. It is also worth noting that our model is trained without using any scenes from NVOS or SPIn-NeRF. In Figure 4, SAM2-Video fails to capture the heads of the T-Rex and the trucks, suggesting that the model struggles to segment complete objects when their colors are similar to the background. In contrast, our MV-SAM produces reliable object masks by leveraging the underlying 3D structure, which separates the T-Rex and the trucks from the background in the pointmaps and thereby simplifies the segmentation problem.

## 4.3 CONTROL EXPERIMENTS

We conduct control experiments to demonstrate the effectiveness of MV-SAM. Note that our control experiments are performed on the ScanNet++ (Yeshwanth et al., 2023) dataset, which serves as the dataset for both training and evaluation.

**Confidence embeddings.** We assign distinct embeddings to low-confidence points and prompts, enabling the model to recognize which inputs are less reliable as described in Eq. 3. As shown in Table 3, incorporating confidence embeddings yields a $7.7\%p$ improvement, suggesting that, without this guidance, the model becomes susceptible to errors arising from inaccurate geometries of low-confidence points. Furthermore, we investigate the effect of varying the proportion of low-confidence points considered, with detailed results provided in Table 13 of the Appendix.

**3D positional embeddings.** In Table 3a, we compare our 3D positional embeddings against the model variants using 2D positional embedding or not using any positional embedding. Without

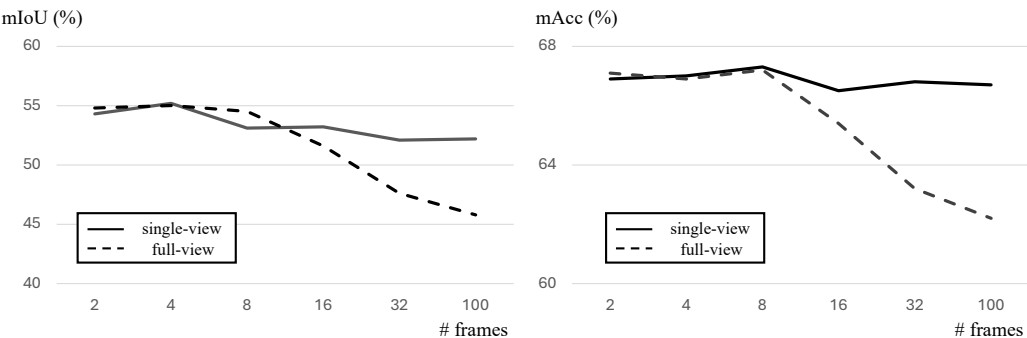

Figure 5: Comparison of single-view attention (ours) and full-view attention. While both approaches perform similarly in few-view setups, full-view attention struggles to scale to a larger number of frames, which is common in practice. Detailed results are provided in the Appendix Table 12.

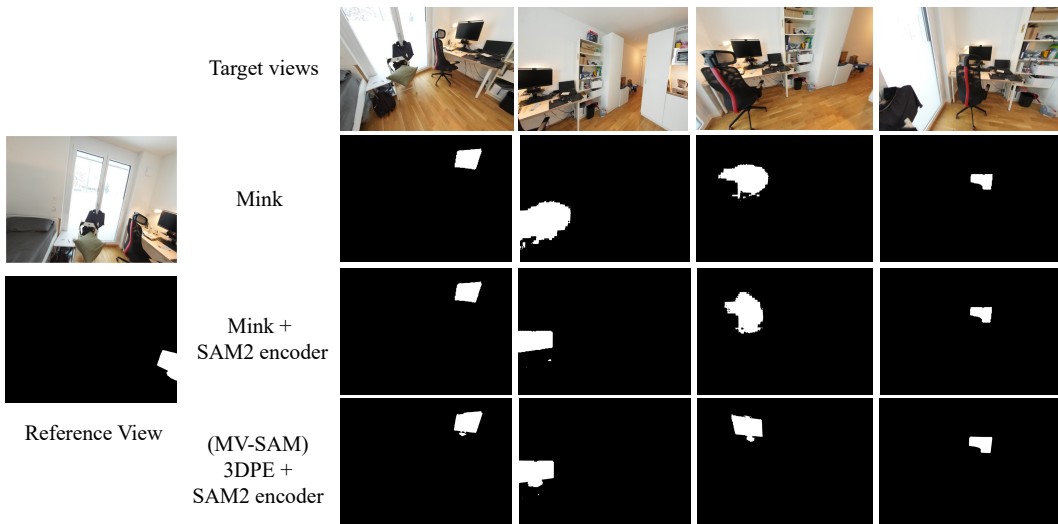

Figure 6: Qualitative results on ablation studies.

positional embeddings, the model fails to localize prompts and often selects incorrect objects during evaluation. With 2D embeddings, where 3D prompts are projected onto individual image planes, the model struggles to determine which prompts remain valid. In this case, some prompts may correspond to occluded regions or even disappear from the scene, making it difficult for the model to reliably identify the intended target. As a result, the model frequently selects occluding objects instead of the actual object of interest, highlighting the inherent limitations of 2D embeddings in handling occlusion.

**Attention scope.** We compare our single-view attention in the mask decoder with full-view attention that aggregates information across all frames, as shown in Table 3 and Figure 5. For a fair comparison, both models are trained with 8 frames per sample. While full-view attention achieves performance comparable to single-view attention when the number of frames is limited to 8 or fewer, its performance degrades substantially as the number of frames increases at evaluation. This degradation stems from the fact that full-view attention introduces a variable number of tokens depending on the frame count, which may require specialized handling for token length extrapolation Press et al. (2022), whereas single-view attention maintains a consistent token structure regardless of frame count. Therefore, we adopt single-view attention, which avoids the instability caused by token length extrapolation.

**Pretrained image encoder from SAM2-Video.** We also compare our MV-SAM against explicit 3D networks that ensure 3D consistency. While keeping the same mask decoder and training losses, we replace our encoder with three variants: (1) a pure 3D encoder using MinkUNet, (2) a pretrained image encoder with voxel-based residual blocks from MinkUNet (Choy et al., 2019), and (3) from PTv3 (Wu et al., 2024). As shown in Table 3b, the pure 3D encoder yields the lowest accuracy, likely

| Model | Dataset info. | Train dataset → Eval dataset | mIoU (↑) | mAcc (↑) |
|---|---|---|---|---|
| MV-SAM (ours) | In-domain data | ScanNet++ → ScanNet++ | 0.510 | 0.694 |
| | Multi-view (small-scale) | uCo3D → ScanNet++ | 0.194 | 0.251 |
| | Single-view (large-scale) | SA-1B → ScanNet++ | 0.489 | 0.635 |
| | In-domain data | uCo3D → uCo3D | 0.910 | 0.965 |
| | Multi-view (small-scale) | ScanNet++ → uCo3D | 0.322 | 0.517 |
| | Single-view (large-scale) | SA-1B → uCo3D | 0.877 | 0.950 |

Table 4: Cross-dataset evaluation results.

because traditional 3D networks assume metric-depthaligned inputs, whereas pointmaps from visual geometry models often exhibit inconsistent scales. Although voxel-based approaches enforce view consistency, their performance is highly sensitive to grid resolution and degrades when pointmap scales vary. Moreover, as shown in Figure 6, the model tends to produce blurry rendering due to their restricted resolution by voxel sizes. In contrast, our model avoids these rigid inductive biases, allowing transformers to implicitly learn 3D consistencyan approach more suitable for pointmaps, which lack metric alignment but preserve rich geometric structure.

**Training on multi-view datasets.** A major challenge in training MV-SAM is the scarcity of multi-view mask-annotated datasets. Although datasets such as ScanNet++ (Yeshwanth et al., 2023) and uCo3D (Liu et al., 2025a) are available, they remain limited in scale, restricted in domain, or biased toward single-object scenes. To assess the effect of dataset choice and scale, we compared MV-SAM trained on the small-scale, multi-view datasets (ScanNet++ and uCo3D) against the single-view, large-scale SA-1B dataset as described in Table 4.

Models trained on ScanNet++ or uCo3D achieve strong performance when evaluated on their corresponding 'in-domain dataset' (e.g., ScanNet++ → ScanNet++ achieving 0.510 mIoU). However, their performance drops substantially when tested in a cross-dataset setting, such as the dramatic drop for uCo3D → ScanNet++ (0.194 mIoU) and ScanNet++ → uCo3D (0.322 mIoU). This confirms that models trained with small-scale, domain-specific multi-view data suffer from poor generalization ability.

In contrast, the model trained on the single-view, large-scale SA-1B dataset demonstrates consistently strong performance across all domains. When evaluating on ScanNet++, the SA-1B model achieved 0.489 mIoU, nearly matching the in-domain ScanNet++ benchmark and dramatically outperforming the uCo3D cross-domain result. Similarly, SA-1B → uCo3D achieves an impressive 0.877 mIoU, almost matching the in-domain uCo3D performance (0.910 mIoU). This evidence highlights the definitive advantage of using a large-scale, diverse dataset like SA-1B as a foundational training resource for MV-SAM, confirming that scale and diversity outweigh multi-view data constraints for achieving robust generalization.

## 5 CONCLUSION

We presented **MV-SAM**, a framework for multi-view promptable segmentation that leverages pointmaps to connect 2D user interactions with multi-view images. By lifting image embeddings from the SAM2-Video's pretrained image encoder into 3D, our method eliminates the need for explicit 3D networks and achieves view-consistent mask prediction without multi-view supervision or 3D annotated dataset. Extensive experiments across NVOS, SpIn-NeRF, ScanNet++, uCo3D, and DL3DV demonstrate that MV-SAM consistently outperforms SAM2-Video and achieves comparable performance with per-scene optimization baselines without any per-scene optimization.

**Limitation.** Since our model relies on an off-the-shelf visual geometry model, its performance is inherently tied to the quality of the pointmaps generated by $\pi^3$. For instance, if $\pi^3$ produces pointmaps with inaccurate depth alignment or structural noise in highly cluttered indoor scenes, these imperfections can propagate to the downstream segmentation process. Furthermore, because our method does not explicitly enforce 3D consistency across views, it may produce unreliable predictions in the presence of outliers, such as misaligned points or artifacts in textureless regions. More description about limitation is included in Appendix H.

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

APPENDIX

# A    IMPLEMENTATION DETAILS

We train our model on SA-1B (Kirillov et al., 2023) dataset that contains 1B images with multiple object masks per image. For each image, we sample 10 prompts, allowing up to 10% to be negative. To supervise dense prompts, we randomly drop 80% of the ground-truth mask and perturb 20% of the initial mask to simulate errors. For focal loss, we set $\alpha = 0.9$ and $\gamma = 1.5$, with loss weights $\lambda_{\text{focal}} = 1.0$ and $\lambda_{\text{DICE}} = 0.05$. As the image encoder, we adopt the publicly available SAM2.1 large checkpoint and keeps frozen during training. Note that we employ the Hiera-L architecture as the visual backbone, initialized with the pre-trained SAM2 checkpoint ('facebook/sam2.1-hiera-large'). Since our framework does not employ an object score predictor not mIoU score predictor, we remove components related to stability and object scores. To suppress sprinkles, we adopt the sprinkle-removal strategy from SAM2-Video, discarding regions whose area is smaller than 0.1% of the total pixels. For evaluation, we select between 10 and 100 frames depending on dataset characteristics.

## A.1    DATASET

**NVOS** dataset (Ren et al., 2022) consists of 8 scenes, each with one reference view and one target view. We follow the official protocol and use the provided scribble prompts. However, the scribble annotation for the *orchid* scene is inaccurate for promptable segmentation. As shown in Figure 7, the positive prompts cover only 3 out of 7 petals. Consequently, SAM2 (Ravi et al., 2024), SA-3D Cen et al. (2023b), SA3D-GS (Cen et al., 2025), and MV-SAM yield low average performance, despite correctly segmenting the corresponding petals. Therefore, we exclude the *orchid* scene from our evaluation and report the per-scene results of previous baselines and MV-SAM in Section F.

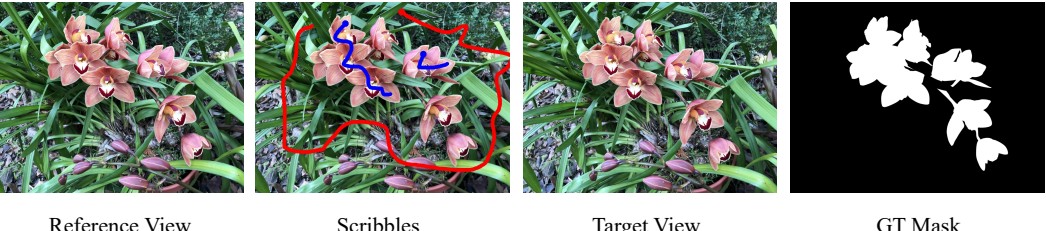

| Reference View | Scribbles | Target View | GT Mask |

Figure 7: Visualization of the *orchid* scene from the NVOS (Ren et al., 2022) dataset.

**SPIn-NeRF** dataset (Mirzaei et al., 2023) provides multi-view masks for target objects. Following the evaluation protocol of SAGA (Cen et al., 2023a), we sample all point prompts from the first viewpoint, sorted by file name. Since the *pinecone* scene is no longer available from the original data source, it is excluded from our evaluation. To facilitate future research, we report the per-scene results, including those of previous baselines, in Section F.

**ScanNet++** dataset (Yeshwanth et al., 2023) provides 3D instance masks but does not include 2D-level instance annotations. The official repository of ScanNet++ offers 2D instance masks rendered via rasterization, which we adopt for evaluating SAM2-Video and MV-SAM. Because ScanNet++ contains objects that occupy only a very small portion of the image, we restrict evaluation to objects covering at least 0.1% of the total image pixels. To ensure coverage of entire scenes, we uniformly sample 100 DSLR frames per scene for evaluation. For training and validation, we follow the official ScanNet++ split, and for evaluation, we select 5 objects per validation scene.

**uCo3D** dataset (Liu et al., 2025a) provides segmentation masks for target objects across large-scale video collections. Given the scale of the dataset, we sample 50 representative sequences for our experiments. Since uCo3D primarily consists of object-centric videos and does not include wide-baseline scenarios with large viewpoint variations, we uniformly sample 50 frames from each sequence to ensure consistent spatial coverage while keeping the evaluation computationally feasible. For evaluation, we directly adopt the official masks provided by uCo3D, which allows for a fair comparison with prior work and ensures reproducibility of our results.

**DL3DV** dataset (Ling et al., 2024) contains diverse indoor and outdoor scenes, each involving multiple objects. Since the dataset does not provide object masks for evaluation, we manually annotated the sequences with the assistance of the SAM2 demo. As SAM2 occasionally produced incorrect tracking results, we refined the annotations and re-propagated the masks until reliable masks were obtained. Once accurate masks were prepared, we uniformly sampled 100 frames from each sequence, resulting in a total of 5 evaluation samples from DL3DV. To ensure transparency and reproducibility, we will release the annotated data together with our code.

## A.2 MODEL

### A.2.1 PRELIMINARIES: $\pi^3$

We employ $\pi^3$ (Wang et al., 2025c) as the visual geometry model of MV-SAM. Here, we provide a more detailed explanation of $\pi^3$ and its improvements over VGGT (Wang et al., 2025a). VGGT is a feed-forward visual geometry transformer that jointly predicts depth, camera poses, and pointmaps from multiple frames. A pointmap refers to a dense per-pixel 3D representation in which each image pixel is mapped to a corresponding 3D point in the world coordinate system. VGGT architecture consists of a DINO-based (Oquab et al., 2023) image tokenization module, a cross-view fusion transformer, and several prediction heads for geometric outputs. While effective, VGGT exhibits a critical limitation: its predictions are sensitive to the order of input views. This permutation sensitivity arises primarily from sequence-dependent positional encodings and asymmetric cross-view attention, both of which treat the set of input images as an ordered sequence rather than an unordered set. As a result, permuting the input frames leads to inconsistent geometric predictions, limiting the robustness of VGGT in the multi-view setup.

To resolve this limitation, $\pi^3$ introduces a permutation-equivariant reformulation of VGGTs multi-view interaction mechanism, ensuring that the predicted geometry remains consistent under any permutation of input frames. Specifically, $\pi^3$ replaces view-index positional encodings with set-based view embeddings that do not encode sequential order, ensuring that each view is treated as an unordered element of a set. Additionally, $\pi^3$ adopts a symmetric cross-view attention module in which all views attend to one another with equal structural weight, eliminating the directional bias inherent in VGGTs attention layers. Furthermore, $\pi^3$ incorporates a permutation-invariant geometry aggregation module that fuses multi-view tokens in a manner consistent with set operations. These architectural modifications guarantee strict permutation equivariance.

With these modifications, $\pi^3$ not only resolves the permutation sensitivity observed in VGGT but also substantially improves the quality of geometric predictions. The resulting consistency across different view orderings leads to more stable depth and pointmap estimation, ultimately enabling state-of-the-art reconstruction performance on both 3D and 4D benchmarks. By adopting $\pi^3$ as its visual geometry model, MV-SAM also inherits permutation equivariance with respect to frame ordering.

### A.2.2 DIFFERENCE BETWEEN 'SINGLE-VIEW' AND 'FULL-VIEW' ATTENTION

In Table 3a, we compare different positional embeddings under various attention strategies. Here, we provide additional details on how the single-view and full-view attention mechanisms operate. The key distinction lies in the scope of cross-attention between image embeddings and prompt tokens. In the single-view setting, cross-attention is computed only between the prompts and the image embeddings of a single reference view. In contrast, the full-view setting broadens this interaction by allowing prompts to attend to the image embeddings from all available views. For clarity, we additionally provide a PyTorch-like code snippet illustrating both attention modules in Listing 1.

### A.2.3 REDUCING MULTI-VIEW PREDICTION TO SINGLE-VIEW TRAINING

As illustrated in Figure 8 (left), during inference, our architecture processes each view independently at the mask-decoding stage, while the visual geometry model still receives all views jointly. Upon analyzing the behavior of MV-SAM, we observed that $\pi^3$ produces highly consistent predictions whether frames are processed individually or as a set. This consistency allows us to approximate the multi-view prediction task using single-view predictions that process all viewpoints independently. Consequently, MV-SAM can be trained with only single-view supervision, yet still generalizes nat-

```python
import torch
import torch.nn as nn
from einops import repeat
from jaxtyping import Float

# Transformer network
transformer = nn.Transformer(..., batch_first=True)

# Data
image_embeds: Float[torch.Tensor, "n_views h w d"]
prompt_embeds: Float[torch.Tensor, "n_prompts d"]  # assume click prompt.

# Method 1: single-view attention (i.e., view-wise attention)
query = repeat(prompt_embeds, "n_prompts d -> n_views n_prompts d")
key = image_embeds.reshape(n_views, h * w, d)
out = transformer(query, key)
out = out.reshape(n_views, h, w, d)

# Method 2: full-view attention
query = prompt_embeds.reshape(1, n_prompts, d)
key = image_embeds.reshape(1, n_views * h * w, d)
out = transformer(query, key)
out = out.reshape(n_views, h, w, d)
```

Listing 1: Comparison between the single-view and full-view attention modules. The single-view attention has a sequence length of $h \times w$, indicating attention is computed for each individual image. In contrast, the full-view attention uses a sequence length of $n\_views \times h \times w$, allowing attention to be computed across all different images. Although our mask decoder employs the two-way transformers detailed in SAM2-Video, we have replaced this component with a simple $nn.Transformer(\cdots)$ in the illustration for ease of comprehension.

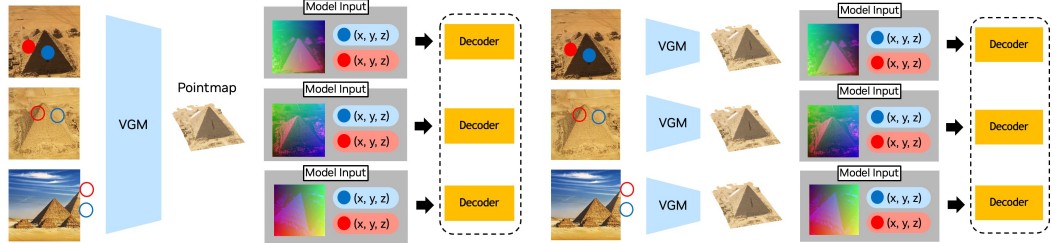

Figure 8: Illustration of how multi-view prediction can be reduced to a set of independent single-view prediction problems. During inference, VGM processes all views jointly (left), whereas during training, each view is handled independently (right). Since the VGM produces nearly identical geometric outputs whether frames are processed jointly or individually, the multi-view prediction task can be effectively reformulated as a single-view prediction problem.

urally to multi-view prediction at inference time thanks to the stable geometric outputs from $\pi^3$.

## B    COMPARISON ACROSS DIFFERENT VISUAL GEOMETRY MODELS

Throughout the MV-SAM experiments, we used $\pi^3$ (Wang et al., 2025c) as our default visual geometry model (VGM), as it provides strong geometric priors and preserves equivariance under frame permutations. To assess how different VGMs influence MV-SAMs performance, we compared three VGMs$\pi^3$, VGGT (Wang et al., 2025a), and WorldMirror (Liu et al., 2025b)under an identical training setup. We follow the evaluation in the main paper on DL3DV experiments. As reported in Table 5, both $\pi^3$ and WorldMirror yield clear performance improvements over VGGT. As reported in WorldMirror, WorldMirror achieves reconstruction quality comparable to $\pi^3$ across various benchmarks while consistently outperforming VGGT. Our results follow the same tendency: MV-SAM

exhibits consistent performance gains when paired with either $\pi^3$ or WorldMirror, compared to VGGT. These findings highlight that the capability of the underlying visual geometry model directly affects MV-SAMs final prediction performance. Therefore, MV-SAM has the potential to be further enhanced with the development of stronger visual geometry models.

Table 5: Performance change of using different VGMs

|  | mIoU (%) | mAcc (%) |
|---|---|---|
| VGGT  (Wang et al., 2025a) | 61.1 | 90.4 |
| WorldMirror  (Liu et al., 2025b) | 74.3 | 92.6 |
| $\pi^3$  (Wang et al., 2025c) | **75.1** | **91.8** |

We conduct a control experiment to evaluate the robustness of MV-SAM against noise in the reconstructed pointmaps generated by visual geometry models. Specifically, we perturb the pointmaps by adding Gaussian noise of varying magnitudes and assess the resulting performance degradation on ScanNet++. The noise scale is defined relative to the standard deviation of the pointmap coordinates; a scale of 1.0 corresponds to adding Gaussian noise with standard deviation to each coordinate.

As shown in Table 6, MV-SAM remains stable under moderate noise levels and maintains strong performance up to a noise scale of 0.5. Interestingly, even under extremely large perturbationssuch as a noise scale of 4.0the model still retains the ability to detect objects by leveraging the strong image embeddings provided by the pre-trained SAM2 encoder. Nevertheless, as expected, the overall performance gradually degrades as the noise magnitude increases.

Table 6: Performance change when injecting noises on predicted pointmaps.

| noise scale $\sigma$ | mIoU (%) | mAcc (%) |
|---|---|---|
| 0.0 | 48.9 | 63.5 |
| 0.25 | 48.4 | 64.2 |
| 0.5 | 47.1 | 65.1 |
| 1.0 | 41.5 | 59.2 |
| 2.0 | 39.7 | 58.4 |
| 4.0 | 33.1 | 49.2 |

## C MODEL STATISTICS

We report the statistics for running MV-SAM and compare them against various baselines-SAGA (Cen et al., 2023a), OmniSeg3D (Ying et al., 2024), and SAM2-Video (Ravi et al., 2024). For a fair comparison, we use 20 frames and measure the pre-processing time for each scene and the inference time for each query. In detail, the pre-processing time for per-scene optimization corresponds to the per-scene training time, whereas for training-free approaches such as SAM2-Video and MV-SAM, it corresponds to their feed-forward computation. As shown in Table 7, per-scene optimization approaches require substantial training time to preprocess features for each scene, while training-free approaches enable fast pre-processing due to their feed-forward predictions.

Nevertheless, per-scene optimization approaches achieve faster inference, as they pre-extract features specific to the scene, whereas training-free approaches require longer inference time. We also observe that our model achieves faster inference than SAM2-Video, since MV-SAM processes all views simultaneously, while SAM2-Video requires iterative mask predictions, which incur linear complexity with respect to the number of frames.

Additionally, we compare the number of 'trainable' parameters and required FLOPs between SAM2-Video and MV-SAM. Since both MV-SAM and SAM2-Video freeze their encoders during training, we only compare the number of learnable parameters. We use the same setup as in Table 7. As shown in Table 8, MV-SAM requires more FLOPs due to the heavy computation of visual geometry models (VGMs). However, MV-SAM does not have the memory modules proposed in SAM2-Video, so ours use less number of trainable parameters than those in SAM2-Video.

Table 7: Training and inference time on a DL3DV scene with 20 frames.

|  | Pre-processing | Inference |
| --- | --- | --- |
| SAGA (Cen et al., 2023a) | 31 (min) | 528 (ms) |
| OmniSeg3D (Ying et al., 2024) | 37 (min) | 463 (ms) |
| SAM2-Video (Ravi et al., 2024) | 3.2 (s) | 4.8 (s) |
| MV-SAM | 5.1 (s) | 1.1 (s) |

Table 8: The number of parameters and FLOPs of SAM2-Video and MV-SAM

|  | # parameters | FLOPs (TFLOPs) |
| --- | --- | --- |
| SAM2-Video (Ravi et al., 2024) | 12.3M | 16.8 |
| MV-SAM (ours) | 4.1M | 44.6 |

## D  ADDITIONAL BASELINES

We additionally compare more recent baselines, SAM2-Long (Ding et al., 2024) and SAM3 (Carion et al., 2025), with MV-SAM with the identical setup in Table 1. Following our evaluation protocol, we benchmarked both methods on three datasets: ScanNet++, uCo3D, and DL3DV. Additionally, we included a simple baseline that unprojects user prompts into 3D and re-projects them into each image to run SAM2 image predictor Ravi et al. (2024) independently per view.

Across all benchmarks, our MV-SAM consistently outperforms the aforementioned baselines, even though these methods are trained on large-scale annotated video datasets. This demonstrates the benefit of explicitly incorporating 3D awareness into the model. Moreover, naively projecting prompts into all views introduces significant challenges in occluded scenes; on ScanNet++, in particular, we observe substantial performance degradation due to the frequent presence of occlusions. Finally, we note that even the latest approach, SAM3, still exhibits inconsistent tracking, despite being trained on complex datasets designed to enhance SAM2.

## E  ADDITIONAL EXPERIMENTS

In this section, we list additional experiments to check our model's performance. We follow the same setup with the control experiments in our main paper.

**Loss for training**  In Table 10, we compare our model using different loss functions, including ASL (BenBaruch et al., 2020), binary cross entropy (BCE), and focal loss (Lin et al., 2017), both with and without the addition of Dice loss. To ensure fairness, all hyperparameters are kept identical across loss configurations.

**3D Decoder**  Another approach is to employ 3D networks as the mask decoder. This design introduces a 3D inductive bias, encouraging neighboring points to produce similar logits. We observed that the grid size strongly affects the final performance, but due to memory constraints, the finest feasible resolution was limited to 0.005. Consistent with the observation in the main paper, voxelization enforces 3D consistency; however, the lack of metric alignment prevents the model from generalizing well across diverse scenes as shown in Table 11.

**Different numbers of frames.**  As discussed in the main paper, our single-view attention outperforms full-view attention, which involves all viewpoints during the attention operations. Below, we provide the exact numerical values corresponding to the graph in Figure 5 in Table 12.

**Confidence-aware Prompting**  We further investigate the proportion of points that should be treated as low-confidence. By varying the low-confidence thresholds, we compare mIoU and mAcc on ScanNet++. We observe that discarding more than 15% of the points leads to performance degra-

Table 9: Comparison of mIoU and mAcc on the ScanNet++, uCo3D, and DL3DV benchmarks. Note that Prompt Projection[†] is a simple baseline that unprojects user prompts into 3D and re-projects those prompts into each image to run the SAM2 image predictor independently per view.

| Method | ScanNet++ | | uCo3D | | DL3DV | |
|---|---|---|---|---|---|---|
| | mIoU (↑) | mAcc (↑) | mIoU (↑) | mAcc (↑) | mIoU (↑) | mAcc (↑) |
| Prompt Projection[†] | 0.292 | 0.592 | 0.782 | 0.833 | 0.412 | 0.702 |
| SAM2-Video (Ravi et al., 2024) | 0.461 | 0.614 | 0.819 | 0.913 | 0.673 | 0.829 |
| SAM2-Long (Ding et al., 2024) | 0.415 | 0.614 | 0.729 | 0.864 | 0.605 | 0.785 |
| SAM3 (Carion et al., 2025) | 0.486 | 0.634 | 0.824 | 0.914 | 0.681 | 0.826 |
| **MV-SAM (Ours)** | **0.489** | **0.635** | **0.877** | **0.950** | **0.751** | **0.918** |

Table 10: Comparison of different loss functions.

| Loss type | mIoU (%) (↑) | mAcc (%) (↑) |
|---|---|---|
| ASL (BenBaruch et al., 2020) | 43.3 | **67.7** |
| ASL (BenBaruch et al., 2020) + Dice | 39.4 | 67.2 |
| BCE | 44.0 | 62.4 |
| BCE + Dice | 49.3 | 64.5 |
| Focal (Lin et al., 2017) | 47.1 | 66.1 |
| Focal (Lin et al., 2017) + Dice (Ours) | **52.2** | 66.7 |

Table 11: Ablation results with different 3D networks and grid resolutions.

| Mask Decoder | Grid | mIoU (↑) | mAcc (↑) |
|---|---|---|---|
| Minkowski | 0.05 | 44.4 | 61.4 |
| | 0.01 | 46.1 | 60.2 |
| | 0.005 | 46.2 | 60.3 |
| PTv3 | 0.05 | 44.1 | 62.2 |
| | 0.01 | 44.2 | 61.0 |
| | 0.005 | 45.4 | 62.4 |
| Ours | - | **52.2** | **66.7** |

Table 12: Performance comparison across different numbers of frames.

| # frames | 2 | 4 | 8 | 16 | 32 | 100 |
|---|---|---|---|---|---|---|
| Single-view (Ours) | 54.3 | 55.2 | 53.1 | 53.2 | 52.1 | 52.2 |
| Full-view | 54.8 | 55.0 | 54.5 | 51.6 | 47.6 | 45.8 |

dation, indicating that removing too many reliable points can negatively impact the results as shown in Table 13.

**Effect of standardization** We apply standardization before extracting sinusoidal embeddings. Experiments show that this strategy improves robustness to different scenarios, such as an increasing number of frames. As shown in Table 14, our model consistently outperforms its variant without standardization. Without standardization, we observe many failure cases, particularly in outdoor scenes. We conjecture that this is because outdoor scenes contain more widely distributed points, which makes the model more prone to inferring outliers during evaluation.

Table 13: Control experiments by varying the confidence threshold.

| Confidence Threshold | mIoU (%) (↑) | mAcc (%) (↑) |
|:---:|:---:|:---:|
| 0 | 44.45 | 61.11 |
| 0.05 | 51.65 | 66.53 |
| 0.10 | 52.18 | **66.81** |
| 0.15 | **52.25** | 66.70 |
| 0.20 | 50.43 | 66.45 |
| 0.25 | 49.55 | 66.18 |
| 0.30 | 49.82 | 66.46 |

Table 14: Effect of using standardization.

| Standardization | w/ (Ours) | w/o |
|:---:|:---:|:---:|
| ScanNet++ | **49.8 / 63.6** | 27.3 / 51.2 |
| uCo3D | **86.9 / 94.4** | 16.2 / 24.5 |
| DL3DV | **35.7 / 64.7** | 6.2 / 13.3 |

## F    PER-SCENE RESULTS FOR NVOS AND SPIN-NERF

Since we excluded the *orchid* scene from NVOS and the *pinecone* scene from SPIn-NeRF, we provide per-scene results for SAM2, MV-SAM, and prior baselines Mirzaei et al. (2023); Cen et al. (2023b; 2025; 2023a); Ying et al. (2024) to facilitate future research. Per-scene mIoU and mAcc are reported in Table 15 for NVOS and in Table 16.

| Method | fern | flower | fortress | horns_center | horns_left | leaves | trex | Avg. |
|---|---|---|---|---|---|---|---|---|
| SA3D (Cen et al., 2023b) | 82.90 | 94.60 | 98.30 | 96.20 | 90.20 | 93.20 | 81.99 | 91.06 |
|  | 94.39 | 98.74 | 99.68 | 99.33 | 99.36 | 99.57 | 97.41 | 98.35 |
| SA3D-GS (Cen et al., 2025) | 85.50 | 96.79 | 98.07 | 98.18 | 94.33 | 93.67 | 82.13 | 92.67 |
|  | 95.30 | 99.21 | 99.64 | 99.52 | 99.55 | 99.60 | 97.47 | 98.51 |
| SAGA (Cen et al., 2023a) | 83.53 | 96.62 | 98.16 | 98.06 | 93.59 | 93.51 | 80.81 | 92.57 |
|  | 94.60 | 99.17 | 99.65 | 99.50 | 99.51 | 99.59 | 97.26 | 98.55 |
| OmniSeg3D (Ying et al., 2024) | 82.70 | 95.30 | 98.50 | 97.70 | 95.60 | 92.70 | 87.40 | 92.84 |
|  | 94.30 | 98.90 | 99.70 | 99.60 | 99.70 | 99.50 | 98.30 | 98.57 |
| SAM2 (Ravi et al., 2024) | 82.83 | 95.20 | 97.03 | 95.71 | 94.66 | 93.37 | 62.24 | 88.72 |
|  | 93.94 | 97.75 | 98.52 | 97.93 | 95.80 | 96.84 | 81.66 | 94.63 |
| MV-SAM | 82.90 | 95.50 | 97.50 | 97.60 | 94.50 | 94.30 | 82.50 | 92.11 |
|  | 94.90 | 98.30 | 98.70 | 97.50 | 98.40 | 98.50 | 95.90 | 97.46 |

Table 15: Per-scene quantitative results on NVOS. The first row of each method corresponds to metric mIoU and the second row to metric mAcc.

## G    QUALITATIVE RESULTS

In this section, we present additional qualitative results. We provide an additional toy experiment to assess whether MV-SAM can recover whole objects when the reference image contains only partial observations. For clearer analysis, we explicitly crop the reference view to enforce scenarios in which the object is only partially visible. We then visualize the corresponding predictions on multiple target views captured from diverse viewpoints. As shown in Figure 9, MV-SAM reliably reconstructs complete object masks even under incomplete reference cues, facilitated by its 3D-aware model design.    Figure 10 shows scene-level pointmaps and their corresponding predicted masks, highlighting the effectiveness of MV-SAM in large-scale sequences. Figures 11, 12, and 13 provide further visualizations on NVOS, MVSeg, and uCo3D datasets, respectively. Lastly, we

| Scene | SPIn-NeRF | SA3D | SA3D-GS | SAGA | OmniSeg3D | SAM-Video | Ours |
|---|---|---|---|---|---|---|---|
| room | 95.6 / 99.4 | 88.22 / 98.33 | 93.73 / 99.18 | 96.91 / 99.59 | 97.9 / 99.7 | 91.90 / 96.00 | 91.50 / 95.90 |
| orchids | 92.7 / 98.8 | 83.55 / 96.86 | 84.68 / 97.18 | 90.55 / 98.29 | 92.3 / 98.7 | 86.90 / 94.32 | 83.50 / 95.70 |
| horns | 92.8 / 98.7 | 94.49 / 99.02 | 95.26 / 99.17 | 92.96 / 98.71 | 91.5 / 98.5 | 84.44 / 92.39 | 89.60 / 95.20 |
| fern | 94.3 / 99.2 | 97.05 / 99.59 | 96.67 / 99.54 | 96.49 / 99.51 | 97.5 / 99.7 | 97.33 / 98.88 | 97.40 / 99.00 |
| fortress | 97.7 / 99.7 | 98.33 / 99.75 | 98.06 / 99.71 | 96.16 / 99.41 | 97.9 / 99.7 | 69.49 / 84.76 | 98.10 / 99.10 |
| leaves | 94.9 / 99.7 | 97.18 / 99.85 | 97.20 / 99.85 | 95.52 / 99.75 | 96.0 / 99.8 | 96.76 / 98.53 | 95.80 / 98.10 |
| fork | 87.9 / 99.5 | 89.41 / 99.55 | 87.91 / 99.49 | 85.84 / 99.42 | 90.4 / 99.6 | 83.88 / 92.67 | 90.00 / 96.00 |
| truck | 85.2 / 95.1 | 90.82 / 96.66 | 94.80 / 98.20 | 95.71 / 98.53 | 96.1 / 98.7 | 83.49 / 91.86 | 96.40 / 98.60 |
| lego | 74.9 / 99.2 | 92.15 / 99.75 | 91.89 / 99.75 | 93.17 / 99.79 | 90.8 / 99.7 | 85.25 / 93.01 | 89.90 / 96.90 |
| Avg. | 90.67 / 98.81 | 92.36 / 98.82 | 93.36 / 99.12 | 93.70 / 99.22 | 94.49 / 99.34 | 86.60 / 93.60 | 92.47 / 97.17 |

Table 16: Per-scene comparison of mIoU / mAcc across different methods on the SPIn-NeRF dataset. Average (Avg.) values are shown in the last row.

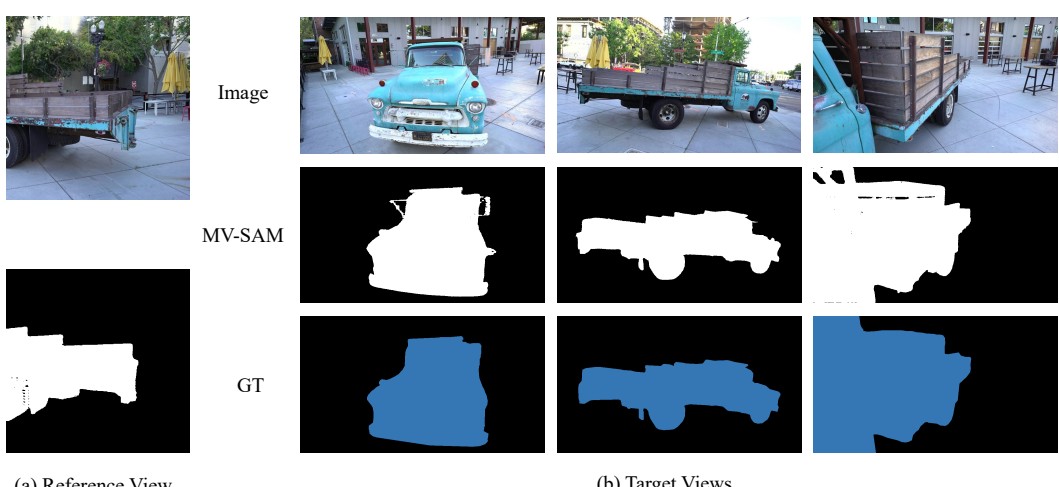

(a) Reference View        (b) Target Views

Figure 9: Visualization of predicted results of MV-SAM for partially occluded cases. We explicitly crop the reference view to encorce scenarios in which the object is only partially observed. MV-SAM reliably reconstructs the complete object masks despite the incomplete reference cues due to the 3D awareness.

include several video results in the supplementary materials, where MV-SAM consistently demonstrates superior performance compared to SAM2-Video.

## H  LIMITATION

Our model also encounters difficulties in more challenging scenarios, such as scenes containing a large number of dynamic objects that break the assumption of static geometry. Similarly, in non-3D-based domains like cartoon animations or synthetic imagery without a consistent underlying geometry, the model struggles to maintain accurate segmentation.

Nevertheless, we observe that our approach can still produce high-quality masks even when the pointmaps are imperfect, as illustrated in Figure 14. For instance, in ScanNet++ scenes with reflective surfaces or partial occlusions, the resulting pointmaps may contain substantial noise. Despite this, our model reliably reconstructs object masks that remain well aligned with the underlying scene structure. This demonstrates a notable degree of robustness to imperfect geometric priors. Further enhancing this robustnesseither by incorporating strategies that explicitly model uncertainty or by leveraging improved pointmap estimation methodsrepresents a promising future work direction.

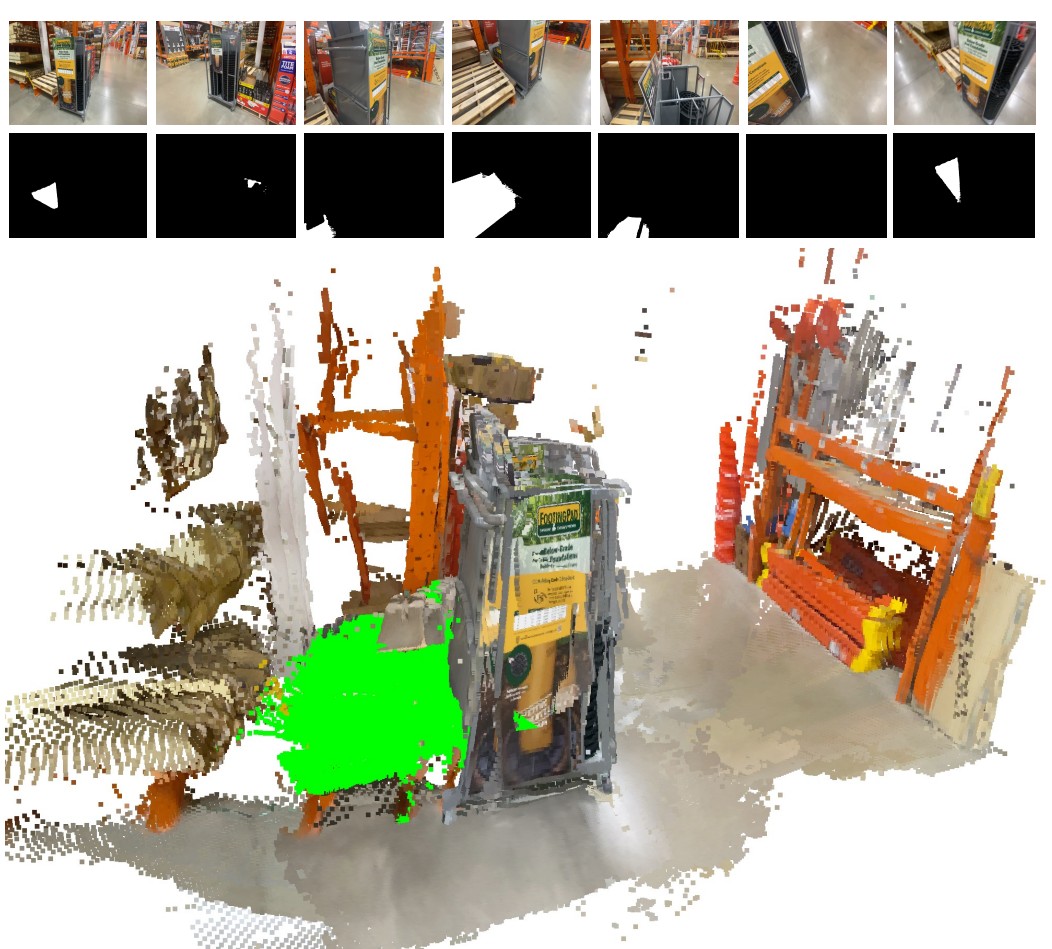

Figure 10: Magnified qualitative results on a DL3DV scene. Owing to the large scale of the scene, detailed predictions are difficult to show in the main paper, so we present magnified results here. The highlighted region in green demonstrates that our method correctly identifies the wood block in the scene.

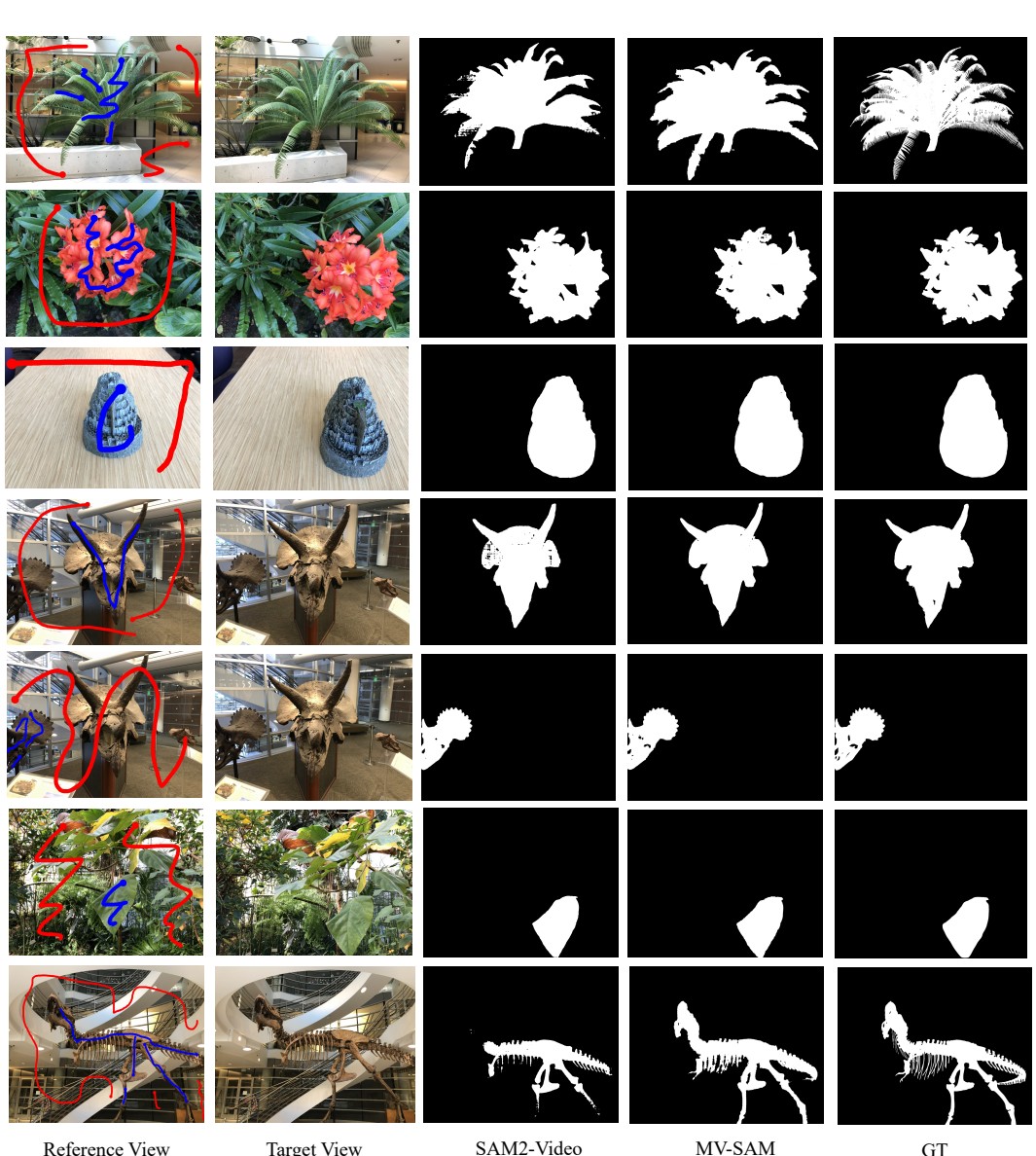

Figure 11: Visualization of predicted masks of SAM2-Video and MV-SAM from the NVOS dataset.

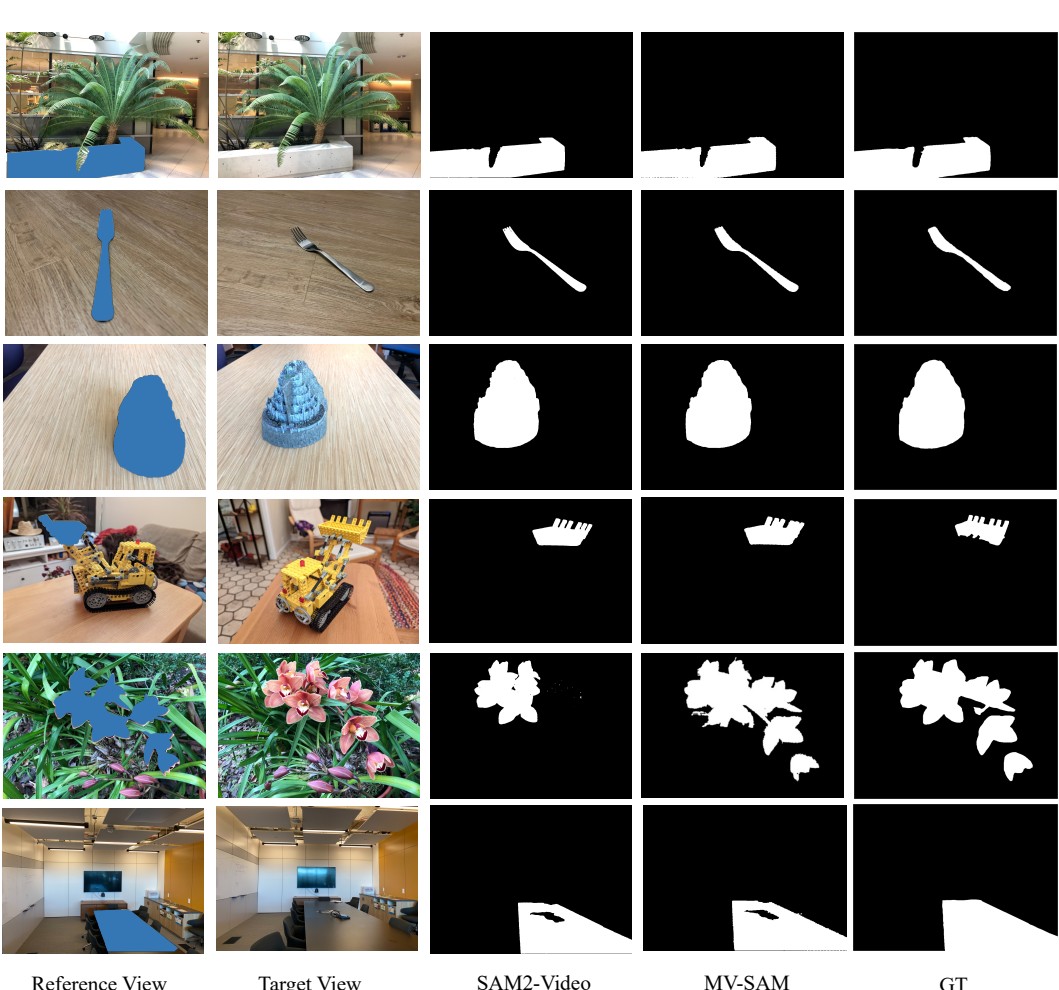

|  Reference View | Target View | SAM2-Video | MV-SAM | GT |

Figure 12: Visualization of predicted masks of SAM2-Video and MV-SAM from the SPIn-NeRF dataset.

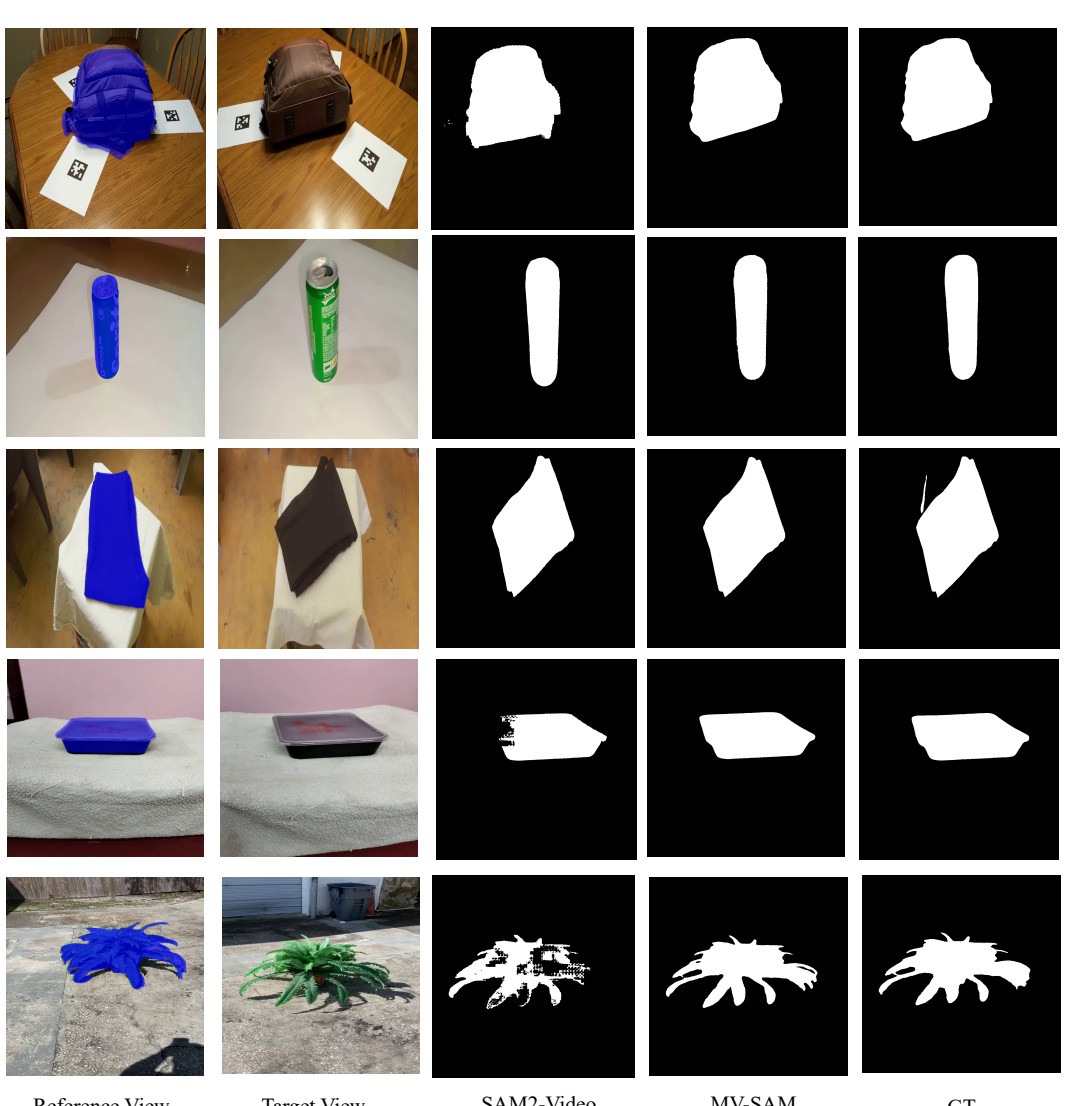

Reference View     Target View     SAM2-Video     MV-SAM     GT

Figure 13: Visualization of predicted masks of SAM2-Video and MV-SAM from the uCo3D dataset.

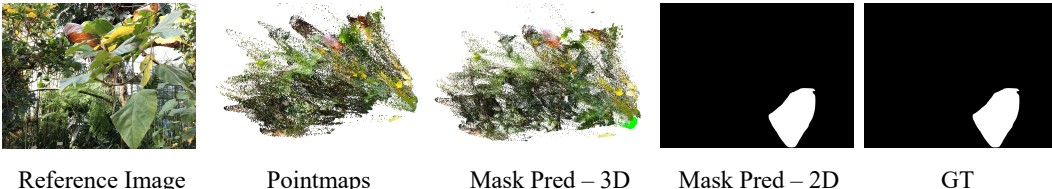

Reference Image     Pointmaps     Mask Pred – 3D     Mask Pred – 2D     GT

Figure 14: Failure cases, where the predicted 2D masks remain clear, but the 3D geometry estimated by $\pi^3$ is inaccurate.

