# OpenReview forum: "MV-SAM: Multi-view Promptable Segmentation using Pointmap Guidance"
_ICLR.cc/2026/Conference — Submitted to ICLR 2026_

### Official Review · Reviewer_zxfL · 2025-10-31

**Soundness:** 3
**Presentation:** 3
**Contribution:** 1
**Rating:** 2
**Confidence:** 4

**Summary:**

Promptable segmentation becomes a more and more popular way for object cutout from images and videos. This work aims to promtable segmentation for multi-view images. More specifically, the user can only provide a prompt, like a click, on one of the views, then we hope to output the mask on all views. How to reach view consistency is the primary challenge. To address this, this work leverage the recent VGGT to obtain a point cloud first, then transfer the 2D segmentation features of SAM into 3D space to produce view-consistent results.

**Strengths:**

- The motivation is clear and method designs make sense

**Weaknesses:**

- My major concern is the limited noverlty. From Fig 2, it seems the key difference is just the modify 2D postional encoder to be 3D version with the help of VGGT. The major contribution comes from existing VGGT.
- It also lacks comparison with a simple baseline：Project the prompt into point cloud and project it onto other views, and do SAM for each view.

**Questions:**

No.

---

> ### Author Response · Authors · 2025-11-26
> **Response to zxfL**
>
> Note that we have uploaded the shared responses for all reviewers in the thread above.
>
> **[zxfL, W1] Limited novelty of proposed method**
>
>
> We appreciate the reviewer for raising this point. Although MV-SAM uses visual geometry models to obtain pointmaps, our contribution goes well beyond simply combining these models with SAM. The key novelty is a framework that turns 2D promptable segmentation into a 3D-consistent multi-view system without requiring any 3D supervision (**Ff9Y**), explicit 3D networks (**Ff9Y**), or per-scene optimization (nx6A). Moreover, the reviewer **5RcK** acknowledges our effort to choose a single-view dataset, SA-1B dataset, for training our multi-view mask prediction pipeline.
>
> Unlike ours, prior work does not address how to lift 2D SAM embeddings into 3D using pointmaps, how to align features across views through 3D-aware cross-attention, or how to achieve consistent multi-view masks using geometry alone. To our knowledge, this design is new, and its effectiveness is demonstrated across five benchmarks where existing 2D video-based approaches show clear limitations.
>
> **[zxfL, W2] Additional baselines**
>
> We identified two recent works—SAM2-Long (Ding et al., 2024) and SAM3 (Carion et al., 2025)—that extend SAM2 (Ravi et al., 2024) to improve long-range object tracking. Following our evaluation protocol, we benchmarked both methods on three datasets: ScanNet++, uCo3D, and DL3DV. Additionally, as suggested by reviewer **zxfL**, we included a baseline that unprojects user prompts into 3D and re-projects them into each image to run SAM2 independently per view. The results are summarized in Table 9.
>
> Across all benchmarks, our MV-SAM consistently outperforms the aforementioned baselines, even though these methods are trained on large-scale annotated video datasets.
> This demonstrates the benefit of explicitly incorporating 3D awareness into the model.
> Moreover, naively projecting prompts into all views introduces significant challenges in occluded scenes; on ScanNet++, in particular, we observe substantial performance degradation due to the frequent presence of occlusions. Moreover, we notice that even the latest approach, SAM3 (Carion et al., 2025), still exhibits inconsistent tracking, despite being trained on complex datasets designed to enhance SAM2.
>
> For more details, please refer to Appendix D.
>
> | Method                  | ScanNet++ mIoU ↑ | ScanNet++ mAcc ↑ | uCo3D mIoU ↑ | uCo3D mAcc ↑ | DL3DV mIoU ↑ | DL3DV mAcc ↑ |
> |-------------------------|------------------|------------------|--------------|--------------|--------------|--------------|
> | Prompt Projection      | 0.292            | 0.592            | 0.782        | 0.833        | 0.412        | 0.702        |
> | SAM2-Video              | 0.461            | 0.614            | 0.819        | 0.913        | 0.673        | 0.829        |
> | SAM2-Long               | 0.415            | 0.614            | 0.729        | 0.864        | 0.605        | 0.785        |
> | SAM3                    | 0.486            | 0.634            | 0.824        | 0.914        | 0.681        | 0.826        |
> | **MV-SAM (Ours)**       | **0.489**        | **0.635**        | **0.877**    | **0.950**    | **0.751**    | **0.918**    |
>
> **Table 9. Comparison of mIoU and mAcc on ScanNet++, uCo3D, and DL3DV.**

---

> > ### Comment · Reviewer_zxfL · 2025-11-27
> > **Thanks for the additional results**
> >
> > Many thanks for your efforts to provide the additional results to compare with the "projection" baselines, where the explanation makes sense to me. Thus I am happy to raise my score.
> >
> > However, it is still very hard to convince me about the novelty. I totally agree that the proposed method is new since it is the first trial to leverage VGGT for MV-SAM. And, I also believe it can reach SOTA. But, it indeed lacks insights because I don't think using vggt to solve MV-SAM is a challenging task. Thus, I raise my score from 2 to 4, but still think the novelty is not good enough for a top paper.

---

### Official Review · Reviewer_5RcK · 2025-10-31

**Soundness:** 3
**Presentation:** 3
**Contribution:** 3
**Rating:** 8
**Confidence:** 3

**Summary:**

This paper presents MV-SAM, a model for multi-view promptable segmentation. The objective is to take a set of unposed images with segmentation prompts (points, boxes, scribbles) for a subset of those images, and output the segmentation mask of the prompted objects for all of the unposed images. While it is possible to apply video-based promptable segmentation models to this task, these models lack explicit 3D understanding, which leads to inconsistent predictions across images. To solve this, MV-SAM uses explicit 3D information from pre-trained 3D reconstruction model $\pi^3$. Specifically, it uses $\pi^3$ to obtain 3D point coordinates for all pixels of the unposed images, and obtains sinusoidal positional embeddings from these 3D coordinates for each pixel and for each prompt, yielding per-pixel 3D positional embeddings and 3D prompts. The pixel-level 3D positional embeddings are then added to extracted image features, and fed to a mask decoder together with the 3D prompts, which finally outputs a segmentation mask for each prompt for each input image. With experiments, MV-SAM is shown outperform video-based promptable segmentation method SAM2 across various datasets.

**Strengths:**

1. The paper presents an original and well-motivated idea. By leveraging the power of 3D reconstruction methods and obtaining a 3D point coordinate for each pixel, it is possible to make the segmentation model aware of the 3D structure of the scene, enabling consistency between the segmentation masks predicted for different images of the same scene.

2. The effectiveness of the proposed method that leverages this idea, MV-SAM, is properly demonstrated through experiments. Across various datasets, MV-SAM outperforms video-based promptable segmentation method SAM2 (see Tab. 1 and Tab. 2), validating the effectiveness of using explicit 3D information. MV-SAM is also shown to perform almost on par with methods that require optimization per individual scene, while MV-SAM generalizes to these same scenes without having seen them during training and is thus much more useful in practice.

3. Overall, the paper is very well written. This makes it easy to understand the contributions and their significance.

4. With the control experiments in Sec. 4.3, the paper properly evaluates the effectiveness of individual contributions and components. For instance, it shows that using confidence embeddings to inform the model of pixels with unconfident 3D reconstruction predictions significantly boosts the performance.

5. A considerable strength of the proposed method is that it can be trained on single-view images, but still performs well when applied to multiple images during inference. This way, it does not depend on multi-view datasets and can be trained on large-scale single-view datasets, for which there is more availability.

**Weaknesses:**

1. This is not a major weakness, but it is not clear what the efficiency is of the proposed MV-SAM method compared to existing method SAM2. I can imagine that running $\pi^3$ for each scene introduces a significant computational overhead. The paper would be stronger if it provided insights into the runtime, number of parameters, and number of FLOPs for both MV-SAM and SAM2. MV-SAM would still be valuable if it were less efficient than SAM2, but information about their relative efficiency would provide insights into the usefulness of each method in practice, and could prompt future research into improving efficiency if necessary.

2. In Sec. 3.3, the paper states:
    > In SAM2-Video, 2D positional embeddings are independently assigned to every frame, making it necessary to use view-wise attention to associate masks and prompts across different views (referred to as ‘full-view’ in Table 3a).

    After also checking the SAM2 paper and architecture, it is not clear what the authors mean by this 'view-wise'  or 'full-view' attention. In what way does the 'single-view' attention by MV-SAM - where there is attention between the features of each individual frame and all prompt embeddings - differ from the attention that is used in the SAM2 mask decoder? This is currently not clear. The paper would be stronger if it better explained - or even better, visualized - what the differences are between these two types of attention, especially because this 'single-view' attention is one of the key components of the mask decoder.

There are some other minor weaknesses, which do not significantly impact my rating:

* There is some inconsistency in Sec. 3.2. Specifically, L240 says that the bottom 15% of the confidence scores are considered 'low-confidence', but L247 says that it is the bottom 20%. Which of the two is correct? This inconsistency should be corrected.
* Typo in L470: 'prdocue' should be 'produce'.

**Questions:**

I believe this is a strong paper, with an original and effective idea and only some minor weaknesses. Therefore, I recommend to accept this paper. Still, there are some things that could be done to improve the paper. Specifically, the paper would be stronger if it included a comparison of the efficiency of MV-SAM and SAM2, if it better explained or visualized what the difference is between the 'single-view' attention by MV-SAM and the 'full-view' attention by SAM2, and if it fixed some minor textual errors. I would recommend the authors to make these changes and provide these explanations in the rebuttal and the revised version.

---

> ### Author Response · Authors · 2025-11-26
> **Response to 5RcK**
>
> Note that we have uploaded the shared responses for all reviewers in the thread above.
>
> **[5RcK, W1] Statistics of SAM2-Video and MV-SAM**
>
> We additionally report the statistics for running MV-SAM and compare them against various baselines - SAGA (Cen et al., 2023a), OmniSeg3D (Ying et al., 2024), and SAM2-Video (Ravi et al., 2024). We report the preprocessing time and inference time of each model on Table 7. In addition, we compare SAM2-Video and MV-SAM with respect to the number of parameters and FLOPs on Appendix C. As shown in Table 8, MV-SAM requires more FLOPs due to the heavy computation of visual geometry models (VGMs). However, MV-SAM does not have the memory modules proposed in SAM2-Video, so ours use less number of trainable parameters than those in SAM2-Video. Please refer to the updated supplementary material.
>
> | Method        | Pre-processing | Inference |
> |---------------|----------------|-----------|
> | SAGA          | 31 (min)       | 528 (ms)  |
> | OmniSeg3D     | 37 (min)       | 463 (ms)  |
> | SAM2-Video    | 3.2 (s)        | 4.8 (s)   |
> | MV-SAM        | 5.1 (s)        | 1.1 (s)   |
>
> **Table 7: Training and inference time on a DL3DV scene with 20 frames.**
>
>
> | Model          | # Parameters | FLOPs (TFLOPs) |
> |----------------|--------------|----------------|
> | SAM2-Video     | 12.3M        | 16.8           |
> | MV-SAM (Ours)  | 4.1M         | 44.6           |
>
> **Table 8: The number of parameters and FLOPs of SAM2-Video and MV-SAM.**
>
>
> **[5RcK, W2] Details of ‘view-wise’ and ‘full-view’ attention**
>
>
> We have added the description of ‘view-wise (=single-view) attention’ and ‘full-view attention’ in Appendix A.2 along with a PyTorch-like code snippet at Listing 1 of the supplementary material. Moreover, we have elaborated how a single-view trained model enables multi-view inference in Appendix A.2.
>
>
>
> **[5RcK, W3] [5RcK, W4]  Writing comments**
>
>
> Thanks for your suggestion. For clarity, we use the confidence threshold of 15%(updated after 5RcK's response to avoid confusion) throughout all experiments. We’ve updated the manuscript to clarify the threshold as 15% and fixed the typo.

---

> ### Comment · Reviewer_5RcK · 2025-11-27
>
> Thank you for your response and for the answers to my questions.
>
> * **Regarding compute statistics:** Thanks for providing the compute statistics. It is valuable that the reader has access to this information. I have one related question, though: I saw that the number of parameters only includes the learnable parameters, thus excluding the frozen encoder. However, I noticed that the paper does not specify what encoder version is used, e.g., Hiera-B or Hiera-L. This information would be useful for reproducibility. Could the authors provide this information?
>
> * **'Full-view' vs. 'single-view' attention:** Thank you for clarifying the difference between single-view and full-view attention. The description and code snippet are clear. However, I am still confused about the statement on L269 & L283-L284, which states that SAM2 uses 'view-wise' attention, which is referred to as 'full-view' attention in Tab. 3a.
>     * This sentence from Sec. 3.3 suggests that 'view-wise' attention is the same as 'full-view' attention, while Listing 1 (appendix) says that 'view-wise' attention is the same as 'single-view' attention. Which of these statements is correct?
>     * Sec. 3.3 states that SAM2 uses 'full-view' attention, and that MV-SAM instead uses the new 'single-view' attention. However, given the description of 'single-view' attention in Appendix A.2.2, 'single-view' attention is essentially the same as the per-frame attention that happens in the mask decoder of the original SAM 2 method, as Fig. 8 of that work [a] shows that the mask decoder only takes the embeddings of a single image as inputs to the attention operation. Could the authors explain why the 'single-view' attention that is a claimed as a new design for MV-SAM is not the same as the attention used in the original SAM 2 mask decoder?
>
>
> [a] Ravi et al., "SAM 2: Segment Anything in Images and Videos," ICLR 2025.
>
> * **Writing comments:** Thanks for processing my feedback. However, there is still an inconsistency. The author response states that a threshold of 20% is used, whereas the updated paper mentions a threshold of 15%. Which of these is correct?
>
> I look forward to your response.
>
> Best, Reviewer 5RcK

---

> ### Author Response · Authors · 2025-11-27
>
> We sincerely thank you for your thorough review and constructive feedback. We have addressed your comments as follows.
>
> **Regarding compute statistics**
>
> Thank you for the suggestion. We have revised the manuscript to explicitly state that our model utilizes the **Hiera-L vision encoder** (L711-713).
>
> **’Full-view’ vs. ‘single-view’ attention**
>
> We acknowledge that our previous descriptions were misleading and potentially confusing. We appreciate your feedback, which has significantly helped improve the clarity of our manuscript. To clearly distinguish between full-view and single-view attention, **we have substantially revised** the relevant paragraph (L269 - L288). To clarify, both SAM2-Video and our MV-SAM utilize the single-view attention scope. The key difference lies in the propagation mechanism: SAM2-Video employs additional memory modules to propagate masks from previous frames, whereas MV-SAM propagates prompts to all frames using pointmaps. We hope this revision resolves your concerns regarding the confusion in the previous manuscript.
>
> **Writing Comments**
>
> We apologize for the confusion regarding the percentage. We confirm that **15%** is the correct value. We have also updated our previous response in this thread to ensure clarity for future readers.

---

### Official Review · Reviewer_nx6A · 2025-11-01

**Soundness:** 2
**Presentation:** 3
**Contribution:** 2
**Rating:** 4
**Confidence:** 3

**Summary:**

This work targets promptable segmentation across multi-view images or videos with 3D consistency.
Prior methods like SAM2-Video lack 3D awareness and produce inconsistent masks across views, while optimization-based methods (SA3D, OmniSeg3D) require costly per-scene fitting.

Their approach uses Pi3 to produce pointmaps (pixel-to-3D correspondences) from unposed images.
The key insight is that pointmaps naturally bridge 2D prompts and 3D geometry without rendering or projection.
Their pipeline
(1) runs Pi3 to get pointmaps with confidences, extracts image embeddings using frozen SAM2-Video.
(2) They then encode the 3d information into positional embeddings for the prompt embedding and their proposed "confidence embedding".
(4) Finally they feed the enhanced embeddings through a mask decoder (standard)

For training, they use only the SA-1B (single-view images) dataset, with no multi-view data or 3D annotations required.
They evaluate on their method on the NVOS and SPIn-NeRF benchmarks, measuring mIoU and mean accuracy for segmentation.
Results show consistent improvement over SAM2-Video (generalization / "online" baseline), while achieving competitive performance with per-scene optimization methods.

**Strengths:**

The technical idea is well-motivated and easy to follow, building off existing work.
Enhancing SAM2-Video embeddings with 3D positional information improves consistency across views.

Results show significant improvement over SAM2-Video across NVOS and SPIn-NeRF benchmarks.
They achieve competitive performance with optimization-based methods without requiring per-scene fitting.

The ablations are informative, particularly Table 3a which evaluates key decoder design choices.
They systematically compare attention scope, positional embeddings, and confidence embeddings.

The Appendix is thorough and I found many questions answered in there.

**Weaknesses:**

The comparison against generalization baselines is limited to SAM2-Video alone.
It would strengthen the paper to include other video or multi-view segmentation methods that don't require per-scene optimization.

The method relies heavily on Pi3 for pointmap generation, but the technical sections provide limited detail on how Pi3 works.
Given that Pi3 appears to do much of the heavy lifting, it's unclear how much of the contribution is genuinely novel versus simply combining existing components (Pi3 + SAM2-Video).
It would be helpful to either describe Pi3 in a preliminary section or compare it against other visual geometry models for generating pointmaps to understand the design choices.

Given that the proposed method slightly underperforms optimization-based methods, it would be useful to show an extra column or so that measures the inference time / number of frames to let their method shine more.

### Minor Weaknesses

The model is trained on single-view object-image pairs from SA-1B (Section 3.4), which seems mismatched with the multi-view/video task at test time.
A bit of justification/analysis on why not just train with multi-view would be helpful.

In general, some of the formatting and the placement of the results could be improved to better help flow for the reader.
This is very minor, but some of the tables are often quite "far" from the reference text.

Table 3b is useful but could be better motivated by introducing some of those methods earlier in the related work.

**Questions:**

Figure 5 compares single-view vs full-view attention, but to my understanding the model was never trained with this full-view attention scenario.
Could the authors elaborate on why this is informative?

Is there any difference between the training/evaluation settings in the ablation studies (Table 3) versus the main results (Table 1)?
I'm trying to make sure I understand the slight difference in results.

---

> ### Author Response · Authors · 2025-11-26
> **Response to nx6A (1 / 2)**
>
> Note that we have uploaded the shared responses for all reviewers in the thread above.
>
> **[nx6A, W1] Additional baselines**
>
> We identified two recent works—SAM2-Long (Ding et al., 2024) and SAM3 (Carion et al., 2025)—that extend SAM2 (Ravi et al., 2024) to improve long-range object tracking. Following our evaluation protocol, we benchmarked both methods on three datasets: ScanNet++, uCo3D, and DL3DV. Additionally, as suggested by reviewer **zxfL**, we included a baseline that unprojects user prompts into 3D and re-projects them into each image to run SAM2 independently per view. The results are summarized in Table 9.
>
> Across all benchmarks, our MV-SAM consistently outperforms the aforementioned baselines, even though these methods are trained on large-scale annotated video datasets.
> This demonstrates the benefit of explicitly incorporating 3D awareness into the model.
> Moreover, naively projecting prompts into all views introduces significant challenges in occluded scenes; on ScanNet++, in particular, we observe substantial performance degradation due to the frequent presence of occlusions. Moreover, we notice that even the latest approach, SAM3 (Carion et al., 2025), still exhibits inconsistent tracking, despite being trained on complex datasets designed to enhance SAM2.
>
> For more details, please refer to Appendix D.
>
> | Method                  | ScanNet++ mIoU ↑ | ScanNet++ mAcc ↑ | uCo3D mIoU ↑ | uCo3D mAcc ↑ | DL3DV mIoU ↑ | DL3DV mAcc ↑ |
> |-------------------------|------------------|------------------|--------------|--------------|--------------|--------------|
> | Prompt Projection†      | 0.292            | 0.592            | 0.782        | 0.833        | 0.412        | 0.702        |
> | SAM2-Video              | 0.461            | 0.614            | 0.819        | 0.913        | 0.673        | 0.829        |
> | SAM2-Long               | 0.415            | 0.614            | 0.729        | 0.864        | 0.605        | 0.785        |
> | SAM3                    | 0.486            | 0.634            | 0.824        | 0.914        | 0.681        | 0.826        |
> | **MV-SAM (Ours)**       | **0.489**        | **0.635**        | **0.877**    | **0.950**    | **0.751**    | **0.918**    |
>
> **Table 9. Comparison of mIoU and mAcc on ScanNet++, uCo3D, and DL3DV.**
>
>
> **[nx6A, W2] Comparison across different visual geometry models**
>
>
> Throughout the MV-SAM experiments, we used Pi3 (Wang et al., 2025c) as our default visual geometry model (VGM), as it provides strong geometric priors and preserves equivariance under frame permutations. To evaluate how different VGMs influence MV-SAM’s performance, we additionally compared three VGMs—Pi3 (Wang et al., 2025c), VGGT (Wang et al., 2025a), and WorldMirror (Liu et al., 2025b)—under an identical training setup. As reported in Table 5, we observe clear improvements over VGGT when using either Pi3 or WorldMirror. According to Table 2 of WorldMirror, WorldMirror achieves on-par reconstruction quality with Pi3 across various benchmarks while showing clear improvements over VGGT. Therefore, our model also shows consistent gains when using either WorldMirror or Pi3 compared to VGGT. This demonstrates that the quality of estimated pointmaps from VGMs affects our prediction performance. For more experimental details, please refer to the Appendix B.
>
> In addition, to make our paper more self-contained, we’ve added a description about VGGT and its extension, Pi3 on Appendix A.2.1.
>
> | VGM                          | mIoU (%) | mAcc (%) |
> |------------------------------|----------|----------|
> | VGGT (Wang et al., 2025)     | 61.1     | 90.4     |
> | WorldMirror (Liu et al., 2025) | 74.3   | 92.6     |
> | π³ (Wang et al., 2025)       | **75.1** | **91.8** |
>
> **Table 5: Performance change of using different VGMs.**

---

> ### Author Response · Authors · 2025-11-26
> **Response to nx6A (2 / 2)**
>
> **[nx6A, W3] Statistics of SAM2-Video and MV-SAM**
>
> We additionally report the statistics for running MV-SAM and compare them against various baselines - SAGA (Cen et al., 2023a), OmniSeg3D (Ying et al., 2024), and SAM2-Video (Ravi et al., 2024). We report the preprocessing time and inference time of each model on Table 7. In addition, we compare SAM2-Video and MV-SAM with respect to the number of parameters and FLOPs on Appendix C. As shown in Table 8, MV-SAM requires more FLOPs due to the heavy computation of visual geometry models (VGMs). However, MV-SAM does not have the memory modules proposed in SAM2-Video, so ours use less number of trainable parameters than those in SAM2-Video. Please refer to the updated supplementary material.
>
> | Method        | Pre-processing | Inference |
> |---------------|----------------|-----------|
> | SAGA          | 31 (min)       | 528 (ms)  |
> | OmniSeg3D     | 37 (min)       | 463 (ms)  |
> | SAM2-Video    | 3.2 (s)        | 4.8 (s)   |
> | MV-SAM        | 5.1 (s)        | 1.1 (s)   |
>
> **Table 7: Training and inference time on a DL3DV scene with 20 frames.**
>
>
> | Model          | # Parameters | FLOPs (TFLOPs) |
> |----------------|--------------|----------------|
> | SAM2-Video     | 12.3M        | 16.8           |
> | MV-SAM (Ours)  | 4.1M         | 44.6           |
>
> **Table 8: The number of parameters and FLOPs of SAM2-Video and MV-SAM.**
>
>
> **[nx6A, W4] Why not train on multi-view?**
>
> We appreciate the reviewer’s comment regarding our use of a single-view dataset (SA-1B) to train a multi-view mask prediction model. Our choice is motivated by the substantial scale imbalance between available datasets. SA-1B contains approximately 1 billion images spanning a wide variety of scenes and object categories, whereas existing multi-view datasets such as DL3DV and ScanNet++ include fewer than 1 million paired images, resulting in an approximate 1,000× difference in scale.
>
> Table 4 further supports this point: although MV-SAM trained on small-scale multi-view datasets performs well on their respective domains, it exhibits limited generalizability to unseen datasets. This demonstrates that the current scale of multi-view datasets is insufficient for training a broadly generalizable multi-view segmentation model.
>
> Based on these findings, we conclude that a large-scale single-view dataset is sufficient for training a model that performs multi-view promptable segmentation. Although certain aspects remain open for future investigation, our experiments consistently support the validity of this choice. We also note that reviewer **5RcK** acknowledged the soundness of using SA-1B as the primary training dataset for our multi-view segmentation framework. We also provide a detailed justification of how single-view training extends to multi-view inference in Appendix A.2.3 and Figure 8.
>
> | Dataset Info.             | Train → Eval          | mIoU ↑ | mAcc ↑ |
> |---------------------------|------------------------|--------|--------|
> | In-domain data            | ScanNet++ → ScanNet++  | 0.510  | 0.694  |
> | Multi-view (small-scale)  | uCo3D → ScanNet++      | 0.194  | 0.251  |
> | Single-view (large-scale) | SA-1B → ScanNet++      | 0.489  | 0.635  |
> |---------------------------|------------------------|--------|--------|
> | In-domain data            | uCo3D → uCo3D           | 0.910  | 0.965  |
> | Multi-view (small-scale)  | ScanNet++ → uCo3D      | 0.322  | 0.517  |
> | Single-view (large-scale) | SA-1B → uCo3D          | 0.877  | 0.950  |
>
> **Table 4: Cross-dataset evaluation results.**
>
>
>
> **[nx6A, W5] [nx6A, W6] Writing comments**
>
> Thanks for your suggestion. We have updated an additional paragraph in Section2 (Related Work) to describe 3D network architecture used in Table 3b.
>
> **[nx6A, Q1] Details of ‘view-wise’ and ‘full-view’ attention**
>
> We have added the description of ‘view-wise (=single-view) attention’ and ‘full-view attention’ in Appendix A.2 along with a PyTorch-like code snippet at Listing 1 of the supplementary material. Moreover, we have elaborated how a single-view trained model enables multi-view inference in Appendix A.2.
>
> **[nx6A, Q2] Performance difference between Table 1 and Table 3**
>
> As noted in L417–L419, we conducted ablation studies on ScanNet++ for both training and inference because training on SA-1B requires substantial computational time, making extensive control experiments impractical. Furthermore, to validate the effectiveness of our single-view training strategy, it was necessary to compare against a multi-view training setup. Since SA-1B provides mask annotations only for single images, we used ScanNet++—which offers multi-view annotations—for both training and inference in the ablation experiments (Table 3). In contrast, the results of the model trained on SA-1B are reported separately in Table 1.

---

### Official Review · Reviewer_Ff9Y · 2025-11-02

**Soundness:** 3
**Presentation:** 3
**Contribution:** 3
**Rating:** 6
**Confidence:** 4

**Summary:**

This paper introduces a method for multi-view promptable segmentation, MV-SAM, using pointmaps based on unposed images using a visual geometry model. The method aims to locate all image regions that correspond to a user prompt in the form of an object mask in a given image. One of the main challenges for this task are 3D awareness and challenges from occlusion, lighting, and textures on the objects. The method creates a pointmap using a pretrained visual geometry model for each image in the given set of unposed images along with image features for each point. This 3D representation maps each pixel to a 3D point along with a confidence map to map the 2D user prompts and 2D image features into a shared 3D space. A transformer-based mask decoder predicts view-consistent masks by attending to the relationship between the 3D image features and the 3D prompts. The model was trained on the SA-1B dataset with single-view object-image pairs using the focal loss and dice loss. Evaluation includes comparisons to state-of-the-art methods such as SAM2-Video and other per-scene optimized methods on diverse real-world datasets covering both indoor and outdoor scenes.

**Strengths:**

Overall, the paper is clear and descriptive about the different components of the proposed method. The problem statement is well-scoped and the method section includes all details and motivations behind the design choices. The core contribution of the method is enabling 3D awareness without 3D supervision by using the pointmap representation. It requires no per-scene optimization allowing for generalization to various datasets. Results show considerable improvement over SAM2-Video and being comparable to per-scene optimized methods. The paper includes ablations for different 3D networks, positional encoding choices, and image encoders to validate the claimed impact of the design choices.

**Weaknesses:**

- The method relies on a pretrained visual geometry model, making it dependent on the accuracy of the pointmap reconstruction. The error in the pointmap reconstruction can propagate to the final mask prediction.
- In Figure 4, I suggest reducing the opacity of the truck in the reference image to make it more visible and easier to interpret.
- The discussion on the limitations is not included in the main text. I recommend that the authors include a discussion with some examples where the method fails. This would help the reader to better understand the contributions of the paper.

**Questions:**

- How well does the model perform if the reference view is occluded but the target view shows the object fully or a different view of the object? I would also like to see the result of the method where the target view is the reference view and the target view is the current reference view with the entire truck visible.
- An analysis on the noise in pointmap reconstruction at inference time would be valuable addition to quantify the dependence on the accuracy of the pointmap reconstruction.

---

> ### Author Response · Authors · 2025-11-26
> **Response to Ff9Y**
>
> Note that we have uploaded the shared responses for all reviewers in the thread above.
>
> **[Ff9Y, W1] [Ff9Y, Q2] Robustness under noises on pointmap**
>
> We additionally conducted a control experiment to evaluate the robustness of MV-SAM against noise in the reconstructed pointmaps generated by visual geometry models. Specifically, we perturb the pointmaps by adding Gaussian noise of varying magnitudes and assess the resulting performance degradation on ScanNet++. The noise scale is defined relative to the standard deviation of the pointmap coordinates; a scale of 1.0 corresponds to adding Gaussian noise with standard deviation to each coordinate.
> We newly add the results at Table 6 of the supplementary material. It shows that MV-SAM remains stable under moderate noise levels and maintains strong performance up to a noise scale of 0.5. Interestingly, even under extremely large perturbations—such as a noise scale of 4.0—the model still retains the ability to detect objects by leveraging the strong image embeddings provided by the pre-trained SAM2 encoder. Nevertheless, as expected, the overall performance gradually degrades as the noise magnitude increases. For more details, please refer to Appendix B.
> | Noise scale | mIoU (%) | mAcc (%) |
> |-----------|----------|----------|
> | 0.0          | 48.9     | 63.5     |
> | 0.25         | 48.4     | 64.2     |
> | 0.5          | 47.1     | 65.1     |
> | 1.0          | 41.5     | 59.2     |
> | 2.0          | 39.7     | 58.4     |
> | 4.0          | 33.1     | 49.2     |
>
> Table 6. Performance change when injecting noises on predicted pointmaps
>
> **[Ff9Y, W2] Reducing the opacity of Figure 4.**
>
>
> Thanks for the suggestion. We have reduced the opacity of the truck in Figure 4 and updated the manuscript.
>
> **[Ff9T, W3] Moving limitations into the main paper.**
>
>
> Following the reviewers’ suggestion, we have relocated the limitation section from the supplementary material to the main paper Section 6. We believe this revision, aligned with your suggestion, will help readers more clearly understand the scope and contributions of our work.
>
> **[Ff9Y, Q1] Robustness to partial observations**
>
>
> We also provide visualizations in Figure 9 showing that MV-SAM correctly recovers the full object in target views even when the reference frame contains only a partially visible instance due to occlusion. Additionally, we include supplementary videos demonstrating that our model reliably predicts the complete object across all views, even when the first frame observes the object only partially.
> In contrast, SAM2-Video often fails to recover the full object because it must implicitly infer that the partially observed region corresponds to the same object in other frames, without access to any 3D cues. MV-SAM, on the other hand, leverages its 3D-aware pointmap representation, allowing it to associate point prompts with the object’s full spatial extent. This results in substantially more reliable tracking under occlusion and partial observation.

---

### Author Response · Authors · 2025-11-26
**Common Reviewer Feedbacks**

We thank all reviewers for their constructive and insightful feedback. We have incorporated the suggested improvements through additional experiments, clarification, and revisions to the manuscript. Below, we summarize the common concerns raised across reviewers and address reviewer-specific comments in separate threads. For brevity, we refer to reviewers using their IDs [Ff9Y, nX6A, 5RcK, zxfL], and denote [W] for weaknesses and [Q] for questions.

As noted by the reviewers, MV-SAM is considered well-written and well-motivated **[Ff9Y, nX6A, 5RcK, zxfL]**, includes informative control experiments **[Ff9Y, nX6A, 5RcK]**, and achieves strong performance **[Ff9Y, nX6A, 5RcK]**. We appreciate these positive assessments. Reviewers also offered several valuable suggestions for further strengthening the work, and we provide our responses to those points below.

We also note that after the submission, we witnessed slight performance gains by simply training for more epochs without modifying any hyperparameters. We have updated the corresponding results in the revised manuscript. In addition, we manually verified mask quality and re-sampled the DL3DV benchmark to ensure a more reliable and fair comparison across models. Consequently, the reported performance of SAM2-Video and MV-SAM has been updated accordingly. We will release the updated checkpoints, code, and data for reproducibility.

---

### Author Response · Authors · 2025-12-04
**Final Discussion Summary**

Dear Area Chairs,

Our MV-SAM introduces a framework for multi-view promptable segmentation that achieves 3D consistency using pointmaps, eliminating the need for explicit 3D networks or per-scene optimization. Our MV-SAM achieves superiority over **SAM 2** and even **SAM 3**. The reviewers have highlighted strengths:

- **Original idea / conceptual novelty** – The method is well-motivated (5RcK) and presents an original idea by leveraging pointmaps to bridge 2D prompts and 3D geometry.
- **Experimental strength** – The method demonstrates considerable improvement over SAM2-Video and achieves competitive performance against per-scene optimization baselines (Ff9Y, nX6A, 5RcK).
- **Practicality** – A key strength is the ability to train on single-view images (SA-1B dataset) while generalizing well to multi-view inference, avoiding reliance on limited multi-view datasets (5RcK).
- **Presentation quality** – The paper is clear and descriptive (Ff9Y). The technical idea is easy to follow (nx6A).


**Reviewer feedback overview**:

- Reviewer 5RcK (Rating 8): Strongly supports acceptance, citing the idea as "original and effective".
- Reviewer Ff9Y (Rating 6): Assessed the paper as good in soundness and contribution.
- Reviewer nX6A (Rating 4): Found the idea well-motivated but requested additional baselines.
- Reviewer zxfL **(Rating updated to 4)**: Explicitly stated "I am happy to raise my score" after we provided the requested "Prompt Projection" baseline and additional results.

We provided comprehensive responses to all reviewers’ comments, and we believe the main concerns have been successfully addressed:

**For Reviewers nX6A and zxfL**:

- Regarding **"additional baselines"**, we benchmarked against recent video methods **SAM2-Long** and **SAM3**, as well as a **"Prompt Projection"** baseline (unprojecting prompts to 3D and re-projecting to views). MV-SAM consistently outperforms all these baselines across ScanNet++, uCo3D, and DL3DV benchmarks.
- Regarding **"novelty"**, we clarified that our framework is the first to enable 3D-consistent multi-view promptable segmentation without 3D supervision, explicit 3D networks, or per-scene optimization, effectively lifting 2D SAM embeddings via 3D-aware cross-attention which is admired by 5RcK as one of our strengths.

**For Reviewer nX6A**:

- Regarding the **"Visual Geometry Model (VGM) comparison"**, we compared our default $\pi^3$ against VGGT1 and WorldMirror3. Results show our model benefits from stronger VGMs, with $\pi^3$ and WorldMirror yielding clear improvements over VGGT.
- Regarding **"single-view training"**, we justified training on SA-1B by showing that models trained on small-scale multi-view datasets (ScanNet++, uCo3D) suffer from poor cross-domain generalization, whereas our single-view trained model generalizes robustly.


**For Reviewer Ff9Y**:
- Regarding **"robustness to noise"**, we added a control experiment adding Gaussian noise to pointmaps. The model remains stable under moderate noise (up to 0.5 scale) and retains detection capabilities even under large perturbations by leveraging strong image embeddings.
- Regarding **"robustness to occlusion"**, we provided qualitative results (Figure 9) and supplementary videos demonstrating that MV-SAM correctly recovers whole objects even when the reference view is partially occluded, unlike SAM2-Video.
- Regarding **"limitations"**, we moved the discussion on limitations from the appendix to the main paper as suggested.

**For Reviewer 5RcK**:
- Regarding **"efficiency statistics"**, we added a comparison of pre-processing/inference time, parameters, and FLOPs. While MV-SAM has higher FLOPs due to the VGM, it requires fewer learnable parameters than SAM2-Video (4.1M vs 12.3M) as it avoids memory modules proposed in SAM2-Video.
- Regarding **"attention terminology"**, we clarified the distinction between "single-view" and "full-view" attention by adding the description and pytorch-style code snippet in Appendix A.2

We thank all reviewers and the Area Chairs for their time, constructive feedback, and thoughtful consideration of our work.

Best regards,

Authors

---

### Meta-Review · Area_Chair_JWsT · 2026-01-07

**Summary:**

This paper gets 1x marginal accept, 1x marginal reject and 1x reject. The strengths of the paper are: 1) 3D awareness without 3D supervision by using the pointmap representation; 2) Results show considerable improvement over SAM2-Video and being comparable to per-scene optimized methods; 3) technical idea is well-motivated and easy to follow; 4) does not depend on multi-view datasets and can be trained on large-scale single-view datasets. The major weaknesses are: 1) limited novelty where key difference is just the modify 2D positional encoder to be 3D version with the help of VGGT. The major contribution comes from existing VGGT. 2) lacks comparison with a simple baseline; 3) comparison against generalization baselines is limited to SAM2-Video alone; 4) method relies heavily on Pi3 for pointmap generation. The major weaknesses on novelty and lack experiments outweigh the strengths of the paper. Furthermore, the average score is leaning negative. The AC thus follows the reviewers comments and scores to reject the paper.

**Reviewer Concerns:**

The major weaknesses are: 1) limited novelty where key difference is just the modify 2D positional encoder to be 3D version with the help of VGGT. The major contribution comes from existing VGGT. 2) lacks comparison with a simple baseline; 3) comparison against generalization baselines is limited to SAM2-Video alone; 4) method relies heavily on Pi3 for pointmap generation.

The AC thinks that all the concerns raised are still outstanding.

**Reviewer Scores:**

The scores would still remain largely the same at 2x positive and 2x negative, and average score leaning negative.

---

### Decision · Program_Chairs · 2026-01-26

Reject